# Learning the non-equilibrium dynamics of Brownian movies

Federico S. Gnesotto[1], Grzegorz Gradziuk[1], Pierre Ronceray [2✉] & Chase P. Broedersz [1,3✉]

Time-lapse microscopy imaging provides direct access to the dynamics of soft and living systems. At mesoscopic scales, such microscopy experiments reveal intrinsic thermal and non-equilibrium fluctuations. These fluctuations, together with measurement noise, pose a challenge for the dynamical analysis of these Brownian movies. Traditionally, methods to analyze such experimental data rely on tracking embedded or endogenous probes. However, it is in general unclear, especially in complex many-body systems, which degrees of freedom are the most informative about their non-equilibrium nature. Here, we introduce an alternative, tracking-free approach that overcomes these difficulties via an unsupervised analysis of the Brownian movie. We develop a dimensional reduction scheme selecting a basis of modes based on dissipation. Subsequently, we learn the non-equilibrium dynamics, thereby estimating the entropy production rate and time-resolved force maps. After benchmarking our method against a minimal model, we illustrate its broader applicability with an example inspired by active biopolymer gels.

[1] Arnold-Sommerfeld-Center for Theoretical Physics and Center for NanoScience, Ludwig-Maximilians-Universität München, D-80333 München, Germany. [2] Center for the Physics of Biological Function, Princeton University, Princeton, NJ 08544, USA. [3] Department of Physics and Astronomy, Vrije Universiteit Amsterdam, Amsterdam, HV 1081, The Netherlands. ✉email: ronceray@princeton.edu; c.broedersz@lmu.de

Over the last two centuries, fundamental insights have been gleaned about the physical properties of biological and soft matter systems by using microscopes to image their dynamics[1,2]. At the micrometer scale and below, however, this dynamics is inherently stochastic, as ever-present thermally driven Brownian fluctuations give rise to short-time displacements[3–6]. This random motion makes such "Brownian movies" appear jiggly and erratic; this randomness is further exacerbated by measurement noise and limited resolution intrinsic to, e.g., fluorescence microscopy[7]. In light of all these sources of uncertainty, how can one best make use of measured Brownian movies of a systems dynamics, to learn the underlying physics of the fluctuating and persistent forces?

In addition to thermal effects, active processes can strongly impact the stochastic dynamics of a system[8–12]. Recently, there has been a growing interest in quantifying and characterizing the non-equilibrium nature of the stochastic dynamics in active soft and living systems[13–25]. In cells, molecular-scale activity, powered for instance by ATP hydrolysis, controls mesoscale non-equilibrium processes in assemblies, such as cilia[26,27], flagella[28], chromosomes[29], protein droplets[30], or cytoskeletal networks[31–34]. The irreversible nature of such non-equilibrium processes can lead to measurable dissipative currents in a phase space of mesoscopic degrees of freedom[9,17,18,35–38]. Such dissipative currents can be quantified by the entropy production rate[39], which is a measure of the irreversibility of the dynamics[40]. New approaches have been developed to measure this rate in real systems[22,24], shedding light onto the structure of dissipative processes[19] and their impact on the dynamics of living matter[20]. However, it remains an outstanding challenge to accurately infer the entropy production rate by analyzing Brownian movies of such systems.

Traditional approaches to measure microscopic forces and analyze time-lapse microscopy data typically rely on tracking the position or shape of well-defined probes, such as tracer beads, fluorescent proteins and filaments, or simply on exploiting the natural contrast of the intracellular medium to obtain such tracks[14–17,29,31,34,41–44]. The tracer trajectories can be studied through stochastic analysis techniques to extract an effective model for their dynamics and infer quantities like the entropy production rate[19,20,22,24,45–48]. There are, however, many cases in which tracking is impractical[49,50], due to limited resolution or simply because there are no recognizable objects to use as tracers. Another, more fundamental limitation of tracking is that one then mostly learns about the dynamics of the tracked object—not of the system as a whole. Indeed, the dissipative power in a system might not couple directly to the tracked variables, and a priori, it might not be clear which coordinates will be most informative about such dissipation. This raises the question how one can identify which degrees of freedom best encode the forces and non-equilibrium dissipation in a given system.

Here we propose an alternative to tracking: learning the dynamics and inferring the entropy production rate directly from the unsupervised analysis of Brownian movies. We first decompose the movie into generic principal modes of motion, and predict which ones are the most likely to encode useful information through a "Dissipative Component Analysis" (DCA). This allows us to perform a dimensional reduction, leading to a representation of the movie as a stochastic trajectory in this component space. Finally, we employ a recently introduced method, Stochastic Force Inference (SFI)[24], to analyze such trajectories. Our approach not only yields an estimate of the entropy production rate of a Brownian movie, which is a controlled lower bound to the system's total entropy production rate, but also important dynamical information such as a time-resolved force map of the imaged system. Thus, our approach may provide an alternative to methods that use microscopic force sensors[43,44,51,52]. In this article, we first present the method in its generality, then benchmark it on a simple two-beads model. Finally, we demonstrate the potential of our approach on simulated semi-realistic fluorescence microscopy movies of out-of-equilibrium biopolymer networks.

## Results

**Principle of the method.** We begin by describing a tracking-free method to infer the dynamical equations of a system from raw image sequences. This approach allows us to determine a bound on the dissipation of a system, as well as the force field in image space.

Our starting point is the assumption that the physical system we observe (Fig. 1a)—such as a cytoskeletal network or a fluctuating membrane—can be described by a configurational state vector $\mathbf{x}(t)$ at time $t$, undergoing steady-state Brownian dynamics in an unspecified $d$-dimensional phase space:

$$\frac{d\mathbf{x}}{dt} = \mathbf{\Phi}(\mathbf{x}) + \sqrt{2\mathbf{D}(\mathbf{x})}\boldsymbol{\xi}(t), \qquad (1)$$

where $\mathbf{\Phi}(\mathbf{x})$ is the drift field, $\mathbf{D}(\mathbf{x})$ is the diffusion tensor field, and throughout this article $\boldsymbol{\xi}(t)$ is a Gaussian white noise vector ($\langle\boldsymbol{\xi}(t)\rangle = 0$ and $\langle\xi_i(t)\xi_j(s)\rangle = \delta_{ij}\delta(t-s)$). Note that when diffusion is state dependent, $\sqrt{2\mathbf{D}(\mathbf{x})}\boldsymbol{\xi}(t)$ is a multiplicative noise term: we employ the Itô convention for the drift, i.e., $\mathbf{\Phi}(\mathbf{x}) = \mathbf{F}(\mathbf{x}) + \nabla \cdot \mathbf{D}(\mathbf{x})$, where $\mathbf{F}(\mathbf{x})$ is the product of the mobility matrix and the physical force in the absence of Brownian noise[53,54].

Our goal is to learn as much as possible about the process described by Eq. (1) from an experimental observation. In particular, we aim to measure if, and how far, the system is

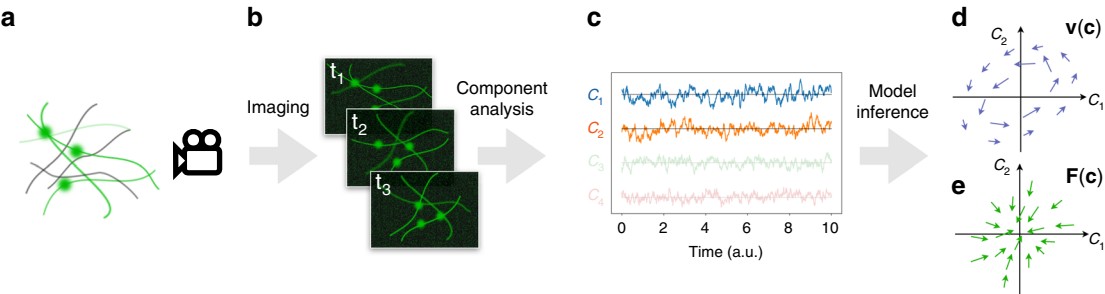

**Fig. 1 Schematic illustration of our approach to learn non-equilibrium dynamics from a Brownian movie. a** Sketch of a network of biopolymers (black) with embedded fluorescent filaments and beads (green). **b** Image-frames of the fluorescent components in panel **a** at three successive timepoints. **c** The time trajectories of the projection coefficients $c_1(t)$, $c_2(t)$, $\cdots$ : the coefficients and respective trajectories discarded by the dimensional reduction are faded. Sketch of the inferred velocity $\mathbf{v}(\mathbf{c})$ (**d**) and of the force field $\mathbf{F}(\mathbf{c})$ (**e**) in the space $\{c_1, c_2\}$.

out-of-equilibrium by determining the irreversible nature of its dynamics. This irreversibility is quantified by the system's entropy production rate[39]

$$\dot{S}_{\text{total}} = \left\langle \mathbf{v}(\mathbf{x}) \boldsymbol{D}^{-1}(\mathbf{x}) \mathbf{v}(\mathbf{x}) \right\rangle, \tag{2}$$

where $\langle \cdot \rangle$ denotes a steady-state average, throughout this article we set Boltzmann's constant $k_{\text{B}} = 1$, and $\mathbf{v}(\mathbf{x})$ is the mean phase-space velocity field quantifying the presence of irreversible currents. Specifically, using the steady-state Fokker–Planck equation one can write $\mathbf{v}(\mathbf{x}) = \mathbf{F}(\mathbf{x}) - \boldsymbol{D}(\mathbf{x})\nabla\log P(\mathbf{x})$, where $P(\mathbf{x})$ is the steady-state probability density function, and flux balance imposes that $\nabla \cdot (P\mathbf{v}) = 0$.

The input of our method consists of a discrete time-series of microscopy images of the physical system $\{\mathcal{I}(t_0), \dots, \mathcal{I}(t_N)\}$—a "Brownian movie" (Fig. 1b). Each image $\mathcal{I}(t)$ is an imperfect representation of the state $\mathbf{x}(t)$ of the physical system as a bitmap, i.e., a $L \times W$ array of real-valued pixel intensities (we neglect the discretization effect induced by the finite number of pixel intensities here). Specifically, we model the imaging apparatus as a noisy nonlinear map $\mathcal{I}(t) = \bar{\mathcal{I}}(\mathbf{x}(t)) + \mathcal{N}(t)$, where $\mathcal{N}$ is a temporally uncorrelated random array representing measurement noise (such as the fluctuations in registered fluorescence intensities), and $\bar{\mathcal{I}}(\mathbf{x})$ is the "ideal image" returned on average by the microscope when the system's state is $\mathbf{x}$. We assume that the map $\mathbf{x} \mapsto \bar{\mathcal{I}}(\mathbf{x})$ is time-independent (i.e., that the microscope settings are fixed and stable).

Importantly, if no information is lost by the imaging process, the ideal image $\bar{\mathcal{I}}(t)$ undergoes a Brownian dynamics equation determined by the nonlinear transformation of Eq. (1) through the map $\mathbf{x} \mapsto \bar{\mathcal{I}}(\mathbf{x})$, as prescribed by Itô's lemma[55]. In general, however, there is information loss and this map is not invertible: due to finite optical resolution or because some elements are simply not visible, the imaging may not capture the full high-dimensional state of the system. For this reason, the dynamics in image space are not uniquely specified by the ideal image value $\bar{\mathcal{I}}$; they also depend on "hidden" degrees of freedom $\mathbf{x}_{\text{h}}$ not captured by the image. In this case, a Markovian dynamical equation for $\bar{\mathcal{I}}$ alone does not exist, but by including the dynamics of $\mathbf{x}_{\text{h}}$, we can write

$$\frac{\mathrm{d}}{\mathrm{d}t}(\bar{\mathcal{I}}, \mathbf{x}_{\text{h}}) = \boldsymbol{\varphi}(\bar{\mathcal{I}}, \mathbf{x}_{\text{h}}) + \sqrt{2\boldsymbol{\mathcal{D}}(\bar{\mathcal{I}}, \mathbf{x}_{\text{h}})}\boldsymbol{\xi}(t). \tag{3}$$

Here $(\bar{\mathcal{I}}, \mathbf{x}_{\text{h}})$ is a column vector composed of pixel intensities $\bar{\mathcal{I}}$ and hidden degrees of freedom $\mathbf{x}_{\text{h}}$, $\boldsymbol{\varphi}(\bar{\mathcal{I}}, \mathbf{x}_{\text{h}})$ and $\boldsymbol{\mathcal{D}}(\bar{\mathcal{I}}, \mathbf{x}_{\text{h}})$ are the drift field and diffusion tensor, respectively, in the combined space of pixel intensities and hidden variables. Our Brownian movie analysis allows us to infer the mean image drift $\boldsymbol{\varphi}(\bar{\mathcal{I}}) := \langle \boldsymbol{\varphi}_{\mathcal{I}}(\bar{\mathcal{I}}, \mathbf{x}_{\text{h}})|\bar{\mathcal{I}} \rangle$ and mean image diffusion tensor $\boldsymbol{\mathcal{D}}(\bar{\mathcal{I}}) := \langle \boldsymbol{\mathcal{D}}_{\mathcal{I}}(\bar{\mathcal{I}}, \mathbf{x}_{\text{h}})|\bar{\mathcal{I}} \rangle$, averaged over the degrees of freedom $\mathbf{x}_{\text{h}}$ lost in the imaging process. From drift and diffusion fields we can directly obtain the mean image-force field $\boldsymbol{\mathcal{F}}(\bar{\mathcal{I}}) = \boldsymbol{\varphi}(\bar{\mathcal{I}}) - \nabla \cdot \boldsymbol{\mathcal{D}}(\bar{\mathcal{I}})$. Similar to force and diffusion fields, the phase-space velocity field $\mathbf{v}(\mathbf{x})$ in the $d$-dimensional physical phase space, transforms into a velocity field $\boldsymbol{\mathcal{V}}(\bar{\mathcal{I}})$ in the $L \times W$-dimensional image space—again, averaged over unobserved degrees of freedom. The corresponding currents result in an apparent entropy production rate associated to the image dynamics,

$$\dot{S}_{\text{apparent}} = \left\langle \boldsymbol{\mathcal{V}}(\bar{\mathcal{I}}) \boldsymbol{\mathcal{D}}^{-1}(\bar{\mathcal{I}}) \boldsymbol{\mathcal{V}}(\bar{\mathcal{I}}) \right\rangle. \tag{4}$$

Importantly, $\dot{S}_{\text{apparent}} \leq \dot{S}_{\text{total}}$: the apparent entropy production rate is a lower bound to the total one. Indeed, all transformations involved in the analysis process—imaging through the nonlinear map $\mathbf{x} \mapsto \bar{\mathcal{I}}(\mathbf{x})$, masking the hidden degrees of freedom, and

averaging over their value—have nonincreasing effects on the entropy production rate (see Supplementary Note 8). The measure of $\dot{S}_{\text{apparent}}$ thus provides direct insight into the dissipative processes in the physical system.

The goal of our method is to reconstruct the mean image-space dynamics $(\boldsymbol{\mathcal{F}}(\bar{\mathcal{I}}), \boldsymbol{\mathcal{D}}(\bar{\mathcal{I}}))$, and in particular the corresponding entropy production rate (Eq. (4)). However, doing so in the high-dimensional image space is unpractical and would require unrealistic amounts of data. We therefore need to reduce the dimensionality of our system to a tractable number of relevant degrees of freedom.

Because each image represents a physical state of the system, we expect that the ideal images $\bar{\mathcal{I}}(t)$ all share similar structural features. Consequently, the Brownian movie occupies only a smaller subspace in the space of all configurations of pixel intensities. To restrict ourselves to the manifold of images representing the physical states and to reduce the noise, we first perform a standard dimensionality reduction procedure: for simplicity, we employ Principal Component Analysis (PCA). As we shall see later, this standard procedure can be reinforced with an analysis that provides an additional basis transformation to select the most dissipative components. The idea behind this approach is to find an appropriate basis, in which pairs of components can be hierarchically ordered according to how much they are expected to contribute to the total entropy production rate. It then becomes possible to truncate the basis and reduce the dimensionality of the problem, while retaining maximum information about the system's irreversible dynamics.

We truncate the basis of components according to two criteria: (1) Noise floor—due to the finite amount of data and the measurement noise present in the Brownian movie, some modes are indistinguishable from the measurement noise. We only keep modes that rise above this noise floor. (2) Time resolution of the dynamics—we only consider the components whose statistical properties are consistent with Brownian dynamics, i.e., such that the short-time diffusive behavior can be resolved through the noise. In low-dimensional systems, these criteria can be extended with an additional restriction based on estimating the dimensionality of the set of images in the Brownian movie.

Our task is now reduced to inferring the mean dynamics in component space,

$$\boldsymbol{\Phi}(\mathbf{c}) := \langle \boldsymbol{\Phi}_{\mathbf{c}}(\mathbf{c}, \mathbf{x}_{\text{h}})|\mathbf{c} \rangle, \quad \boldsymbol{D}(\mathbf{c}) := \langle \boldsymbol{D}_{\mathbf{c}}(\mathbf{c}, \mathbf{x}_{\text{h}})|\mathbf{c} \rangle \tag{5}$$

where $\mathbf{c}(t) = (c_1(t), c_2(t), \dots, c_n(t))$ are the components obtained after a linear transformation of the images (see Fig. 1c), $\boldsymbol{D}_{\mathbf{c}}$ is the restriction of the diffusion tensor to the $\mathbf{c}$-space, and the hidden degrees of freedom $\mathbf{x}_{\text{h}}$ now also include those present in the image, but left out after the components' truncation. This procedure has reduced the system's dynamics to that of a smaller number of components, making it possible to learn $\boldsymbol{\Phi}(\mathbf{c})$ and $\boldsymbol{D}(\mathbf{c})$.

To this end, we employ a recently introduced method, Stochastic Force Inference[24] (SFI), for the inverse Brownian dynamics problem. Briefly, this procedure is based on a least-squares approximation of the diffusion and drift fields using a basis of known functions (such as polynomials). This method is data-efficient, not limited to low-dimensional signals or equilibrium systems, robust against measurement noise, and provides estimates of the inference error, making it well suited for our purpose. In practice, we use SFI in two ways: (1) we infer the velocity field $\mathbf{v}(\mathbf{c})$ (Fig. 1d) and the diffusion field $\boldsymbol{D}(\mathbf{c})$, which we use to measure the entropy production rate. (2) We infer the drift field $\boldsymbol{\Phi}(\mathbf{c})$, compute the image-force $\mathbf{F}(\mathbf{c}) = \boldsymbol{\Phi}(\mathbf{c}) - \nabla \cdot \boldsymbol{D}(\mathbf{c})$ (Fig. 1e), and thus reconstruct the dynamics of the components. To render this deterministic dynamics more intelligible, we can transform $\mathbf{F}(\mathbf{c})$ back into image space by inverting the $\bar{\mathcal{I}} \mapsto \mathbf{c}$

linear transformation: this results in a pixel force map, which indicates at each time step which pixel intensities tend to increase or decrease. This provides, we argue, a way to gain insight into the dynamics of Brownian systems and disentangle deterministic forces from Brownian motion without tracking.

Our analysis framework can thus be schematically summarized as: imaging → component analysis → model inference (Fig. 1). This procedure allows the inference of entropy production rate and reconstruction of the dynamical equations from image sequences of a Brownian system.

**A minimal example: two-beads Brownian movies.** Next, we test the performance of our procedure on a simple non-equilibrium model: two coupled beads moving in one dimension. The beads are coupled by Hookean springs with stiffness $k$ and experience Stokes drag with friction coefficient $\gamma$, due to the surrounding fluid (Fig. 2a). In this two-bead model, the time-evolution of the bead displacements

$\mathbf{x}(t) = (x_1(t), x_2(t))$ obeys the overdamped Langevin Eq. (1), with $\mathbf{F}(\mathbf{x}) = \mathbf{Kx}$ and $K_{ij} = (1 - 3\delta_{ij})k\gamma^{-1}$. The system is driven out of thermodynamic equilibrium by imposing different temperatures on the two beads: $D_{ij} = \delta_{ij}k_B T_i \gamma^{-1}$[9,22,56–58]. First, we obtain position trajectories for the two beads by discretizing their stochastic dynamics using an Euler integration scheme (see Supplementary Note 1). Then, we use these position trajectories to construct a noisy Brownian movie (Fig. 2b) (cf. Supplementary Note 2 and Supplementary Movie 1). Note that by construction, the steady-state dynamics of the two-beads system in image space is governed by a nonlinear Langevin equation with multiplicative noise.

We seek to reduce the dimensionality of the data and to filter out measurement noise by finding relevant components. To this end, we employ Principal Component Analysis (PCA)[59] and determine the basis of $n$ principal components $\mathbf{pc}_1, \mathbf{pc}_2, \ldots, \mathbf{pc}_n$ to expand each image around the time-averaged image $\langle \mathcal{I} \rangle$: $\mathcal{I}(t) = \langle \mathcal{I} \rangle + \sum_{i=1}^{n} c_i(t)\mathbf{pc}_i$. The dynamics of the projection

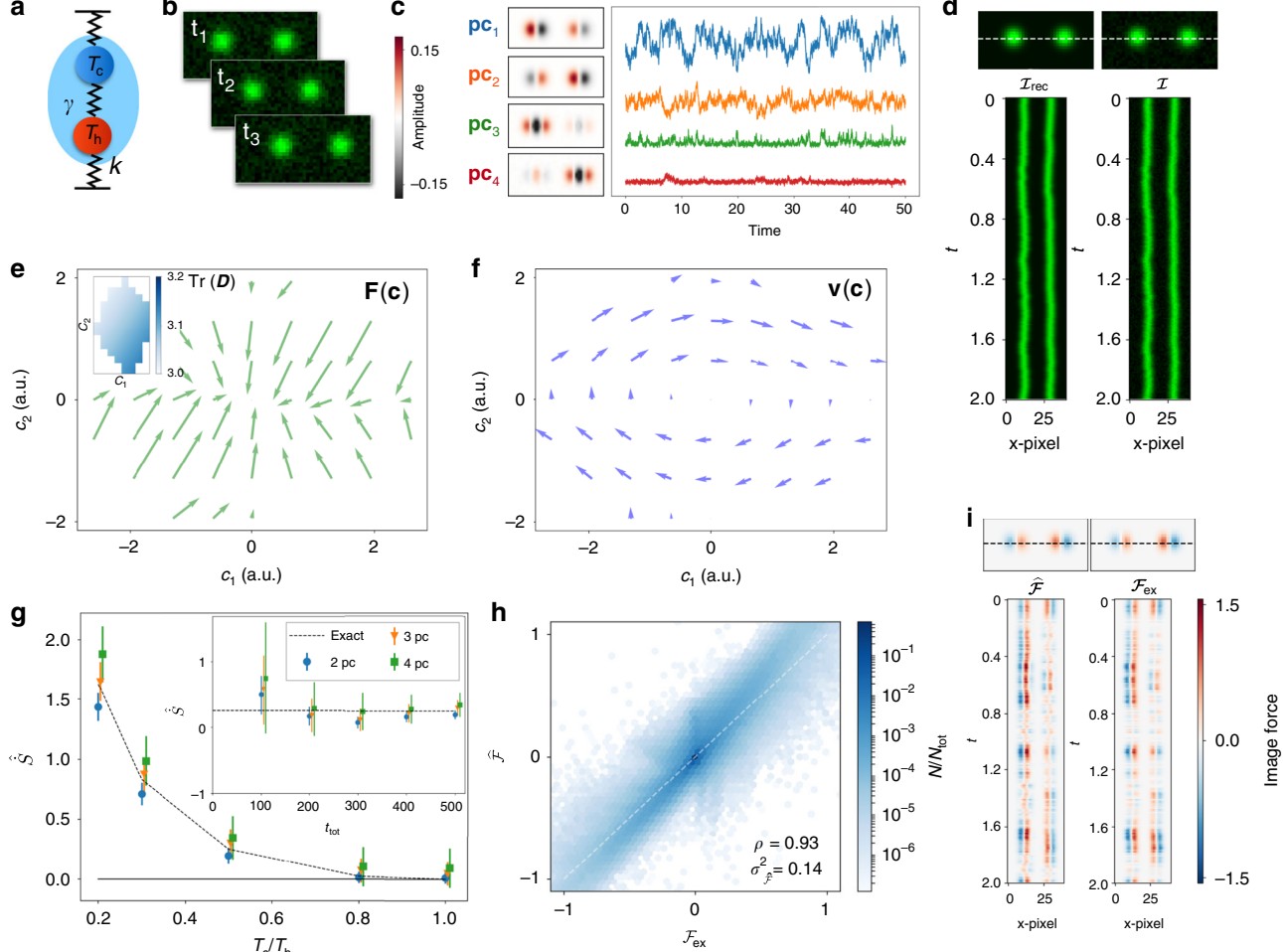

**Fig. 2 Benchmarking the Brownian movie learning approach with a simple toy model. a** Schematic of the two-bead model. The temperature of the hot bead $T_h = 1$ is fixed and the temperature of the cold bead $T_c \leq 1$ is varied. **b** 40 × 20 Frames of the noisy (10% noise) Brownian movie for the two-bead model at successive timepoints. **c** The first four principal components (in arbitrary units) with time traces of respective projection coefficients. The color map displays negative values in black and positive values in red. **d** Snapshot $\mathcal{I}_{rec}$ of the reconstructed movie, reconstructed with the first four principal components, and snapshot $\mathcal{I}$ of the original movie (right), together with associated kymographs. Pixel intensity ranges from 0 (black) to 1 (bright green). We compare pixel intensities along the superimposed horizontal dashed line. Force field (**e**) and mean phase-space velocity (**f**) in the space of the first two principal components $\{c_1, c_2\}$. Arrows are scaled for visualization purposes. Inset **e**: trace of diffusion tensor $\mathrm{Tr}(\mathbf{D})$ with the same axis scaling. **g** Inferred entropy production rate $\hat{S}$ for varying temperature ratio $T_c T_h^{-1}$ and number of included principal components. Inset: $\hat{S}$ as a function of trajectory length for a fixed $T_c T_h^{-1} = 0.5$. The error bars represent an estimate of the root-mean-square deviation between the true apparent entropy production rate and the inferred value (see Methods). **h** Scatter plot of the elements of the exact image-force field $\mathcal{F}_{ex}$ vs. the inferred image-force field $\hat{\mathcal{F}}$ for different pixels and timepoints (data have been binned for visualization purposes). Results are obtained using the first four principal components. **i** Comparison of inferred $\hat{\mathcal{F}}$ and exact $\hat{\mathcal{F}}_{ex}$ image-space force fields, together with associated kymographs.

coefficients are on average governed by the drift field $\boldsymbol{\Phi}(\mathbf{c})$ and diffusion tensor $\boldsymbol{D}(\mathbf{c})$ (see Eq. (5)).

In the simulated data of the two-bead model, the first four principal components satisfy criteria (1) and (2) introduced above (Fig. 2c). Interestingly, $\mathbf{pc}_1$ and $\mathbf{pc}_2$ resemble the in-phase and out-of-phase motion of the two beads, respectively, and should suffice to reproduce the dynamics of $(x_1(t), x_2(t))$. The components $\mathbf{pc}_3$ and $\mathbf{pc}_4$ appear to mostly represent the isolated fluctuations of the hot and cold beads and mainly account for the nonlinear details of the image representation. Together, the first four components allow for an adequate reconstruction of the original images (Fig. 2d, Supplementary Fig. 1).

From the recorded trajectories in $\mathbf{pc}_1 \times \mathbf{pc}_2$ space we can already infer key features of the system's dynamics using SFI. Specifically, we infer the force and diffusion fields (Fig. 2e). In the phase space spanned by the first two principal components, we identify a stable fixed point at $(0, 0)$ (Fig. 2e). As may be expected in this case, the $\mathbf{pc}_1$-direction (in-phase motion) is less stiff than the $\mathbf{pc}_2$ direction (out-of-phase motion).

The temperature difference between the two beads results in phase-space circulation, as revealed by the inferred mean velocity field (Fig. 2f). To quantitatively assess the irreversibility associated with the presence of such phase-space currents, we estimate the entropy production rate of the system $\widehat{S}$, which converges for long enough measurement time (Fig. 2g-inset). Strikingly, already with two principal components we find good agreement between the inferred and the exact entropy production rate, capturing from $78 \pm 25\%$ at $T_c T_h^{-1} = 0.5$ to $88 \pm 7\%$ of the entropy production rate at $T_c T_h^{-1} = 0.2$ (Fig. 2g). Furthermore, the difference between the exact and inferred entropy production rate is consistent with the typical inference error predicted by SFI. As expected, the estimate of the entropy production rate increases with the number of included components. Note that including more modes than the dimension of the physical phase space (in this case 2) can lead to an overestimate of $\dot{S}$ (Fig. 2g). In such low-dimensional systems, one can further restrict the number of included components based on estimating the dimensionality of the set of images in the Brownian movie.

We can also use the information contained in the first four principal components to quantitatively infer forces in image-space via the relation $\widehat{\boldsymbol{\mathcal{F}}}(\boldsymbol{\mathcal{I}}(t)) = \sum_{i=1}^{4} \widehat{F}_i(\mathbf{c}(t)) \mathbf{pc}_i$. Note that while two modes were sufficient to infer $\widehat{S}$, more modes are needed to reconstruct the full images and image-force fields as a linear combination of modes. When inferring forces we always subtract from the drift the spurious force $\nabla \cdot \boldsymbol{D}(\mathbf{c})$ arising in overdamped Itô stochastic differential equations with multiplicative noise[53,54]. For comparison purposes, the exact image-force field is obtained directly from the simulated data as: $\widehat{\boldsymbol{\mathcal{F}}}_{\text{ex}}(t) = \{\bar{\boldsymbol{\mathcal{I}}}[\mathbf{x}(t) + \mathbf{F}(\mathbf{x}(t))\Delta t] - \bar{\boldsymbol{\mathcal{I}}}[\mathbf{x}(t)]\}\Delta t^{-1}$. Remarkably, we find good qualitative agreement between inferred and exact image-force fields for specific realizations of the system, as shown in the kymographs in Fig. 2i (see also Supplementary Movies 2 and 3). Moreover, we find a strong correlation (Pearson correlation coefficient $\rho = 0.93$) between inferred and exact image forces. To further quantify the performance of force inference, we compute the relative squared error on the inferred image-force field $\sigma_{\widehat{\boldsymbol{\mathcal{F}}}}^2 = \sum_t \| \widehat{\boldsymbol{\mathcal{F}}}(t) - \widehat{\boldsymbol{\mathcal{F}}}_{\text{ex}}(t) \|^2 \left( \sum_t \| \widehat{\boldsymbol{\mathcal{F}}}(t) \|^2 \right)^{-1}$, which in this case is modest, $\sigma_{\widehat{\boldsymbol{\mathcal{F}}}}^2 = 0.14$ (Fig. 2h).

Thus, with sufficient information, we can use our approach to accurately predict at any instant of time the physical force fields in image space from the Brownian movie, even if the system is out of equilibrium. Moreover, the results for this simple two-bead system demonstrate the validity of our approach: we reliably infer

the non-equilibrium dynamics of this system. Arguably, direct tracking of the two beads is, in this case, a more straightforward approach. However, this changes when considering more general soft assemblies comprised of many degrees of freedom.

**Dissipative component analysis.** To expand the scope of our approach, we next consider a more complex scenario inspired by cytoskeletal assemblies: a network of elastic filaments (Fig. 3a). The filaments are modeled as Hookean springs represented as bonds connecting neighboring nodes of a triangular network. We randomly remove bonds to introduce spatial disorder in the system. The state of the network as a whole, represented by the set $\{\mathbf{x}_i\}$ of two-dimensional displacement of each node $i$, undergoes Langevin dynamics (Eq. (1)). In this case, the force acting on node $i$ is $\mathbf{F}_i(\mathbf{x}) = -\sum_{j \sim i} \frac{k_{ij}}{\gamma}(\| \mathbf{x}_{i,j}(t) \| - \ell_0)\hat{\mathbf{x}}_{i,j}$, where $k_{ij} = k$ if the bond is present, $k_{ij} = 0$ if it is not, $\mathbf{x}_{i,j} = \mathbf{x}_i - \mathbf{x}_j$, $\hat{\mathbf{x}}_{i,j}$ is the corresponding unit vector, and the sum runs over the nearest-neighbor nodes $j$ of node $i$. Rigid boundary conditions are imposed to avoid rotations and diffusion of the system as a whole. Finally, we drive the system out of equilibrium by randomly setting a fraction of the network nodes at an elevated temperature, as illustrated in Fig. 3a.

To study an experimentally relevant scenario, we generate a Brownian movie of a random filamentous network (Supplementary Note 2), which is only partially imaged (black frame in Fig. 3a) with measurement noise and at a limited optical resolution (Supplementary Note 6, 7). To simulate limited optical resolution, we blur the image-frames of the movie with a Gaussian filter (Fig. 3b and Supplementary Movie 4). In this spatially extended system, generated from an underlying dynamics with 800 degrees of freedom, it is not obvious based on the recorded Brownian movie ($80 \times 80$ pixels) how to select and analyze the relevant degrees of freedom.

We start our movie-based analysis by employing PCA to reduce the dimensionality of the image data (Fig. 3c). For this set of simulation data, our truncation criteria indicate that the maximum number of retainable components is roughly 200 (Supplementary Note 5 and Supplementary Fig. 3). Although we greatly reduced dimensionality of the image data using this truncation, it is still intractable to infer dynamics in a 200-dimensional space due to limited statistics. However, even a subset of these modes may suffice to glean useful information about the system's non-equilibrium dynamics. Therefore, as a first attempt, we infer the dynamics in increasingly larger PC-space via SFI. This allows us to infer the retained percentage of entropy production rate $\widehat{S}/\dot{S}_{\text{ex}}$ in the observed region (See Supplementary Note 2) as a function of the number of principal components considered (Fig. 3e). In contrast to the two-beads case, we observe that in this more realistic scenario we recover less than 4% of the entropy production rate of the observed system with the first 30 PCs. Indeed, PCA is designed to find modes that capture the most variance in the image data, and large variance does not necessarily imply large dissipation. Thus, in this case, PCA fails at selecting components that capture a substantial fraction of the entropy production rate.

Our goal is to infer the system's non-equilibrium dynamics. We thus propose an alternative way of reducing data dimensionality that spotlights the time-irreversible contributions to the dynamics, which we term Dissipative Component Analysis (DCA). DCA represents a principled approach to determine the most dissipative pairs of modes for a linear system with state-independent noise (see Supplementary Note 3). For such a linear system, there exists a set of component pairs for which the entropy production rate can be expressed as a sum of

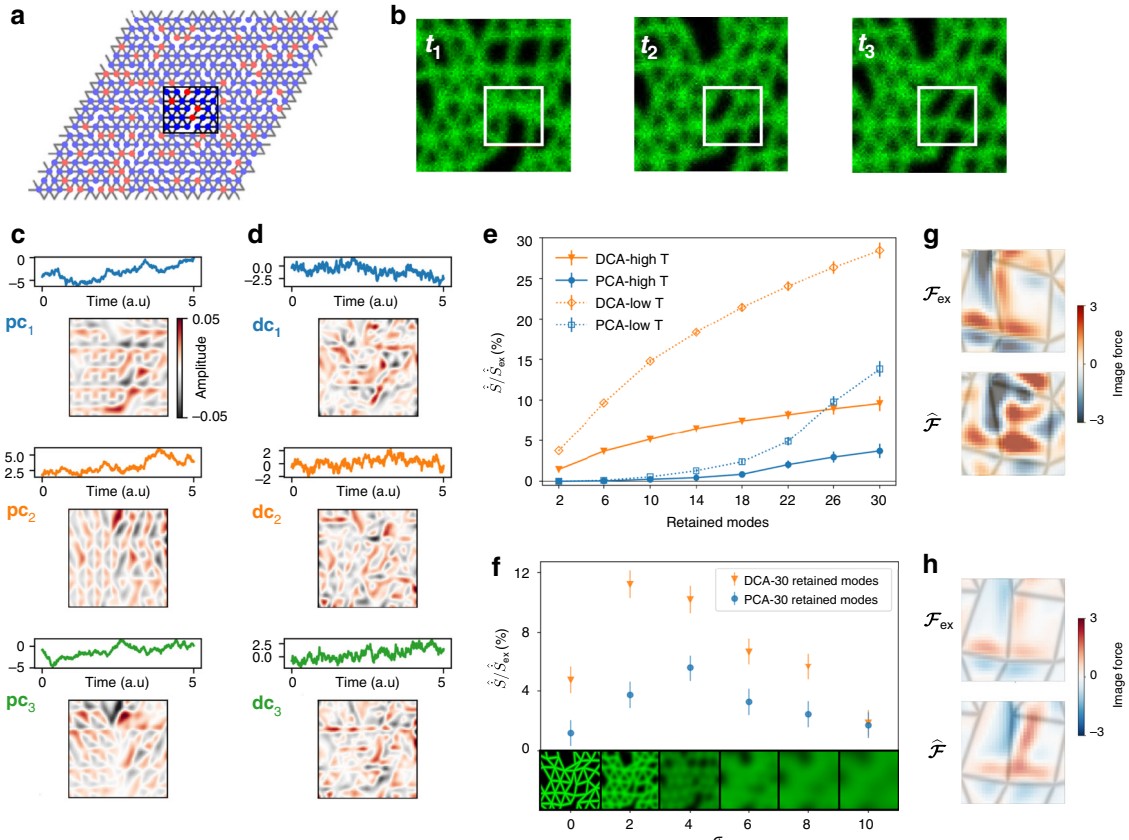

**Fig. 3 Learning the non-equilibrium dynamics of Brownian movies of simulated filamentous networks. a** The 20 × 20 filamentous network generated in the Brownian dynamics simulation with 20% random bond removal and heterogeneous temperatures: node temperatures are randomly set to $T_{hot}$ with probability 0.2, or else to $T_{cold} = 0.2\ T_{hot}$. The black frame indicates the observed region of the system, which is analyzed with our movie-based method. **b** Three time frames of the Brownian movie of the observed region of the system (80 × 80 pixels, $T_{hot} = 0.25$). **c, d** Trajectory of the projection coefficient $c_i$ in arbitrary units together with associated image-component for PCA (**c**) and DCA (**d**) for the observed region defined in panel **a**. Scale bar applies to all image-components. **e** The recovered entropy production rate $\hat{S}/\dot{S}_{ex}$ for the observed region as a function of the number of components included in the analysis. For the high and low temperature cases $T_{hot} = 0.25$ and $T_{hot} = 0.05$, respectively. See Supplementary Note 5 and Supplementary Fig. 2 for additional data at equilibrium and convergence of the estimates with total time. **f** The recovered entropy production rate $\hat{S}/\dot{S}_{ex}$ as a function of the blurring parameter $\sigma$ for 30 retained PCs and DCs. We show a corresponding blurred frame above every x-axis tick. The error bars in panels **e, f** represent an estimate of the root-mean-square deviation between the true apparent entropy production rate and the inferred value (see Methods). **g, h** Comparison of the exact image-force $\mathcal{F}_{ex}$ to the inferred one $\widehat{\mathcal{F}}$ at a selected instant of time for the region of interest in the white frame in panel **b** for the high (**g**) and low (**h**) temperature cases. The underlying network structure is drawn in gray as a guide to the eye.

independent positive-definite contributions, which can be ranked by magnitude. After a suitable truncation, this basis ensures that the components with the largest entropy production rate are selected. While the approach is only rigorous for a linear system with state-independent noise, we demonstrate below that this method also performs well for more general scenarios.

DCA relies on the measurement of an intuitive trajectory-based non-equilibrium quantity: the area enclosing rate (AER) matrix $\dot{A}$ associated to a general set of coordinates $\mathbf{y}$. The elements of the AER matrix, in Itô convention, are defined by[24,38,60–62]

$$\dot{A}_{ij} = \frac{1}{2}\langle y_j \dot{y}_i - y_i \dot{y}_j \rangle, \qquad (6)$$

where $y_i$ denotes the $i$-th coordinate centered around its mean value and $\langle \cdot \rangle$ a time average. This non-equilibrium measure quantifies the average area enclosed by the trajectory in phase space per unit time. Importantly, the AER is tightly linked to the entropy production rate. Specifically, for a linear system $\dot{S} = \mathrm{Tr}(\dot{A}C^{-1}\dot{A}^T D^{-1})$ where the covariance matrix $C_{ij} = \langle y_i y_j \rangle$. DCA identifies a basis of vector pairs $\{(\mathbf{dc}_1, \mathbf{dc}_2); (\mathbf{dc}_3, \mathbf{dc}_4); \ldots\}$ that simultaneously transforms $C$ to the identity and diagonalizes $\dot{A}\dot{A}^T$

(see Supplementary Note 3). By doing so, DCA naturally separates the entropy production rate into independent contributions that can be readily ordered by magnitude, i.e., $\dot{S} = \dot{S}_{\mathbf{dc}_1,\mathbf{dc}_2} + \dot{S}_{\mathbf{dc}_3,\mathbf{dc}_4} + \cdots$ with $\dot{S}_{\mathbf{dc}_1,\mathbf{dc}_2} > \dot{S}_{\mathbf{dc}_3,\mathbf{dc}_4} > \cdots$. Truncating the basis of dissipative components using the aforementioned criteria, allows us to identify a limited number of components that are assured to maximally contribute to the dissipation of the system. This is analogous to PCA, where the diagonalization of the covariance matrix $C$ allows one to select the components which capture most of the variance.

To test the performance of DCA, we revisit the network simulations. We first perform PCA to reduce noise and dimensionality. Subsequently, we perform DCA with the first 200 principal component coefficients as input. The dissipative components exhibit a different spatial structure than the principal components, as they aim to maximize different quantities (Fig. 3d). Strikingly, DCA allows us to recover a larger portion of the entropy production rate of the observed region (almost 10% with 30 components), performing consistently better than the PCA-based approach, as shown in Fig. 3e. Finally, we note that the performance of our approach improves substantially in

systems with smaller fluctuations in which the image-space dynamics is closer to linear (Fig. 3e and Supplementary Movie 8).

In non-equilibrium systems our DCA-based method infers non-zero entropy production rates, even with poor optical resolution (Fig. 3f, Supplementary Note 6, and Supplementary Fig. 4) and with strong measurement noise (Supplementary Note 7 and Supplementary Fig. 5). At the same time we measure no dissipation in equilibrium systems. Thus, this example illustrates the potential applicability of our approach to real experiments on biological assemblies.

Our inference approach reveals additional information about the dynamics in the system, such as force field estimates. These force fields provide insight into the spatial structure of the instantaneous deterministic forces in the system at a given configuration. In image space, these forces describe the dynamics of the pixel: positive and negative image forces represent a deterministic force acting to, respectively, raise and lower pixel values, which reflect the forces acting on the position and shape of the objects being imaged. To investigate to what extent our movie-based learning approach reconstructs the elastic forces exerted by the network's filaments, we exploit the short range of the interactions in the system to facilitate extracting information about local forces from local dynamics in image space. We consider a small region of interest (white frame in Fig. 3b, Supplementary Movie 5) and compare the inferred force field in image space to the exact one. For this purpose, we employ PCA in our dimensional reduction scheme, which can be used both in and out of equilibrium. Inferring image-force fields with high accuracy for this complex example is challenging (Pearson correlation coefficient between exact and inferred images force $\rho = 0.37$ for the high temperature case and $\rho = 0.56$ for the low temperature case). Nonetheless, despite the network disorder, large fluctuations, many hidden degrees of freedom, limited optical resolution, and measurement noise, we find that the inferred force field in image space can capture the basic features of the exact force field, as shown in Fig. 3g, h (Supplementary Movies 6–11). Finally, we emphasize that our approach is scalable: force inference on a small spatial region of interest can be applied to arbitrarily large systems, as long as the interactions are local.

## Discussion

We considered the dynamics of movies of time-lapse microscopy data. Under the assumptions outlined in the first section of the Results, these movies undergo Brownian dynamics in image space: the image-field obeys an overdamped Langevin equation of the form of Eq. (3). Rather than tracking selected degrees of freedom, we propose to analyze the Brownian movie as a whole.

Our approach is based on constructing a reduced set of relevant degrees of freedom to reduce dimensionality, by combining PCA with a new method that we term Dissipative Component Analysis (DCA). In the limit of a linear system with state-independent noise, DCA provides a principled way of constructing and ranking independent dissipative modes. The order at which we truncate is an important trade-off parameter of this method: on the one hand we wish to significantly reduce the dimensionality of the data, on the other hand we need to include enough components to retain the information necessary to infer the system's dynamics. After the dimensional reduction, we infer the stochastic dynamics of the system, revealing the force field, phase-space currents, and the entropy production rate in this basis. This information can then be mapped back to image-space to provide estimators for the stochastic dynamics of the Brownian movie. We illustrated our approach on simulated data of a minimal two-beads model and on complex filamentous networks in both equilibrium and non-equilibrium settings, and showed that it is robust in the presence of measurement noise and with limited optical resolution. Beyond providing controlled lower bounds of the entropy production rates directly from the Brownian movie, our approach yields estimates of the force fields in image space for an instantaneous snapshot of the system and we demonstrated that this approach can be scaled up to large systems. Thus, we provide in principle an alternative to microscopic force and stress sensing methods[43,44,51,52].

We focused here on a class of soft matter systems termed "active viscoelastic solids"[9,63]. Such systems include active biological materials, such as cytoskeletal assemblies[31,33,34,64], membranes[16,65,66], chromosomes[29], protein droplets[30], as well as active turbulent solids[67], and colloidal systems[10]. Although these structures are constantly fluctuating both due to energy-consuming processes (e.g., rapid contractions generated by molecular motors) and thermal motion, they do not exhibit macroscopic flow. Useful insights into the properties of such systems have been obtained via different noninvasive techniques. Typically, these techniques employ time traces of tracked objects to extract information about the active processes governing the non-equilibrium behavior[16–20,58,65]. Often, however, it is not a priori obvious, which physical degrees of freedom should be tracked, how tracking can be performed in fragile environments, and to what extent the dynamical information about the system of interest is encoded in the measured trajectories[49]. While tracking-free approaches have been proposed to obtain rheological information of a system under equilibrium conditions[50], our approach offers an alternative to tracking that can provide information on dissipative modes and the instantaneous force fields of a fluctuating non-equilibrium system.

In summary, we presented a viable alternative to traditional analysis techniques of high-resolution video-microscopy of soft living assemblies. Indeed, we envision experimental scenarios where our approach may serve as a guide, providing insights by disentangling the deterministic and stochastic components of the dynamics, and by helping to identify the source of thermal and active forces as well as the dissipation in the system. Overall, our movie-based approach constitutes an adaptable tool that paves the road for a systematic, noninvasive and tracking-free analysis of time-lapse data of soft and living systems.

## Methods

Parameters for Fig. 2: We use $k = 2$, $\gamma = 1$, $k_B = 1$. Panels c–f, h, i have been obtained with $T_c T_h^{-1} = 0.5$ and for a trajectory of length $t_{tot} = 10^5 \Delta t$, $\Delta t = 0.01$. Panel g with $t_{tot} = 5 \times 10^4 \Delta t$. We employed a first order polynomial basis for the inference of forces and diffusion fields using SFI. The noise-corrected estimator was used to infer the diffusion fields[24].

Parameters for Fig. 3: All results have been obtained with a trajectory of $10^6$ time steps, $\Delta t = 0.005$ and $80 \times 80$ pixels frames for the observed region of the full network. We employed a first order polynomial basis for the inference of forces and diffusion fields using Stochastic Force Inference, and noise-corrected diffusion estimates. The high temperature case is shown in panel g using 50 PCs and the low temperature case is shown in panel h using 20 PCs.

The error bars on the entropy production rates in Figs. 2, 3 are obtained as a self-consistent estimate of $\sqrt{\langle (\dot{s}_{apparent} - \dot{s})^2 \rangle}$, where $\langle \cdot \rangle$ represents the average over the realizations of the noise. For details see ref. [24].

## Data availability

The datasets generated during and/or analysed during the current study are available from the corresponding authors on reasonable request.

## Code availability

The Python code generating the data and implementing the analysis presented in the manuscript is available at https://github.com/ronceray/NonequilibriumBrownianMovies.

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

## Acknowledgements

We thank C. Schmidt, F. Mura, S. Ceolin, and I. Graf for many stimulating discussions. This work was Funded by the Deutsche Forschungsgemeinschaft (DFG, German Research Foundation) under Germany's Excellence Strategy—EXC-2094 - 390783311 and by the DFG grant 418389167.

## Author contributions

P.R. and C.P.B. conceived the project. F.S.G. wrote the new codes developed in this manuscript and performed all simulations and Brownian movies analysis. P.R. provided support for the SFI analysis and G.G. largely developed the derivations underlying the DCA analysis. All authors contributed conceptually to developing the Brownian movie analysis and DCA frameworks, interpreting the results, and writing the paper.

## Funding

## Competing interests

The authors declare no competing interests.
