## [Peer Review File · Nature Communications]

REVIEWERS' COMMENTS:

Reviewer # 1 (Remarks to the Author):

The authors set out to introduce a new data analysis technique for data obtained in microscopy experiments. In many soft matter experiments—especially with living matter—one obtains time-lapse microscopy videos of the systems evolution. Whereas, often people use data obtained from tracking embedded particles, the authors here aim to use the entire video for inference. Their goal is two-fold. One is to infer something about the entropy production of the system and two is to infer aspects of its dynamics.

Their method proceeds by lifting the dynamics to an image space where they estimate entropy production. They improve the data by processing it both using a PCA analysis as well as introduce a novel DCA (dissipation components analysis) to aid with the entropy production estimation. This is because, while the PCA is successful at finding directions where there is the most variation, they are not necessarily those directions that are most dissipative. Thus, the authors introduce another decomposition/projection of the dynamics using insights from linear diffusive processes.

The authors test their method on two examples. Two coupled beads in viscous media at two different temperatures as well as a network of coupled harmonic oscillators with a media of random temperatures. Their method is able to capture a sizable fraction of the true entropy production and in addition does a reasonable job of inferring the underlying force field. Both models are assumed to fulfill their assumption of Brownian dynamics

In many respects, this is a timely work. There has been a lot of interest lately in developing experimentally viable methods to infer entropy production in nonequilibrium matter, especially living systems. This paper brings to bear many of these tools and uses them in an innovative way.

1.1) My main reservation for this paper is whether it has broad applicability. The authors central assumption on which the method rests is that the observed physical system evolves according a multidimensional Brownian dynamics (Eq. 1). It is this assumption that allows them to use the formula in Eq. (2) to estimate the entropy production rate. Generically, one would expect that the degrees of freedom observed in a microscopy experiment on a complex nonequilibrium system are only part of all the dynamics occurring, and that many degrees of freedom are coarse-grained away. Thus, one would expect the observed dynamics to be highly non-linear and non-Markovian. The question then arises is if you attempt to fit the dynamics to a Brownian dynamics (as in Eq. 1) and then infer the entropy production from the fitted dynamics (Eq. 2), does this entropy production have anything to do with true observable entropy production (accounting for non-Markovian and non-Gaussian effects)? It is my understanding that the answer here is false. There is no relation between the entropy production and the entropy production inferred (via Eq. 2) for a fit to a Brownian dynamics. This fact calls into question how principled this approach is, when one cannot be confident that the observed dynamics are well described by Eq. 1. I would suggest that the authors run their analysis on a non-Markovian/non-Gaussian example to test whether their estimated entropy production offers any relation to physical entropy production; and support those conclusions with analytic calculations.

1.2) Minor points:

1) Section III, after \hat{x} there is a stray period in the first paragraph.

2) The authors also might find of interest (J. Phys. Conf. Series 750 (2016) 012003)) where those authors introduce and discuss the notion of probability angular momentum which bears a striking resemblance to the flux loops of the present article.

3) In the author's analysis of the DCA in referring to figure 3d, they write "the dissipative components appear to reflect the local temperature inhomogeneities". Unfortunately, I cannot see this in Fig. 3d. While it's clear that the dissipative components have a weight distribution that is not evenly distributed, how does one make that connection to the variations in temperature?

Reviewer # 2 (Remarks to the Author):

Gnesotto et al. present a new approach for inferring non-equilibrium physical characteristics of systems, such as entropy production rate and time-resolved force distributions, from time-lapse microscopy imaging data. Their approach involves a new dimensionality reduction technique that uses dissipation to select a basis of modes. To test and demonstrate their approach, they use it to analyze the dynamics of two systems, a simple two-bead and spring system and a more complex filament network. The paper is well-written, with clear presentation of methodology and results, and presents a novel analysis approach that avoids the use of tracer particles and particle tracking. Due to the accessibility of the approach and its potential broad application to biophysics and soft-matter physics, the work represents a significant contribution to the suite of techniques for inferring the non-equilibrium physics of living and non-living systems.

While the manuscript is of good quality as is, I do have the following major and minor suggestions for strengthening the the work:

Major

2.1) While the authors demonstrate the effects of noise on their analysis approach, it would be interesting to know how robust the authors' methods are to differences in optical resolution (or similarly, the size of the structure of interest relative to the optical resolution of the imaging system). Structure in biological systems, such as cytoskeletal elements of the cytoplasm, is often near or below the diffraction limit, and given that the manuscript is in part targeting experimentalists, addressing this practicality could strengthen the work. On a related note, is there some rationale for choosing the specific parameters used to generate the Brownian movies (e.g. Gaussian with variance 9 pixels in the case of the two-bead model)?

2.2) How was the patch size for the patching procedure (results in Fig. 3F) chosen? Also, in Supplementary Fig. 4, it looks as though the centers of the patches lie at the vertices of the filament network. How do results depend on patch size and placement with respect to structure in the images? Again, addressing these questions could strengthen the work by accounting for practical considerations.

2.3) What are the reasons that PCA was chosen over other dimensionality reduction techniques. Non-negative matrix factorization, for example, would avoid negative pixel values (e.g. Fig. 2C). There are certainly good reasons to choose PCA, as evidenced by, for example, the methods the authors use to select the appropriate number of components for downstream analysis, but

it would be nice to see some of this spelled out in the text.

2.4) What are the reasons that the 5x5 filamentous network was chosen? In the introduction (last sentence), the authors argue that analysis of this network demonstrates that the approach is scalable to large systems, but I do not know that it is entirely clear given the scale and complexity of, for example, biopolymer networks in living systems. Additional justification or motivation for the particular system parameters could be sufficient to address this.

2.5) Minor

A) Given the potentially broad readership, it could be nice to see a bit more motivation for why entropy production rate is such a quantity of interest.

B) Is there a reference for the claim that entropy production rate quantifies irreversibility (the sentence before Equation 2) and that irreversibility quantifies how far from equilibrium a system is (as implied by the preceding sentence)?

C) Please add axes labels to the plots in Supplementary Fig. 3.

D) Does DCA yield any new information or improvement to analysis of the two-bead system?

E) On page 4, second column, second paragraph, is there a reference for the claim in the sentence beginning “Importantly, when inferring forces we always subtract from the drift...”

F) In Fig. 2d, might there be a better way to illustrate the similarities between the exact and reconstructed images, such as some kind of highlighting of colocalization? It’s a bit hard to visually evaluate similarity as is.

G) Why was the particular model for imaging noise (uniformly sampled white noise) chosen? I suspect the particular noise model should not substantially affect results, but some combination of Gaussian white noise and shot noise (Poisson) might model imaging noise in high resolution microscopy a bit more realistically (see e.g. <https://rupress.org/jcb/article/185/7/1135/35453/Accuracy-and-precision-in-quantitative>).

Overall, the manuscript represents a significant contribution by introducing a new analysis approach that should be broadly of interest to those studying the non-equilibrium dynamics of soft matter and living systems. The clear presentation of results and methodology should facilitate adoption of the new analysis techniques. I would not want to unnecessarily delay publication of the work by burdening the authors with excessive additional work, but I hope that my comments might provide some constructive guidance for possible improvements.

Reviewer # 3 (Remarks to the Author):

“Learning the Non-Equilibrium Dynamics of Brownian Movies”

Federico S. Gnesotto, Grzegorz Gradziuk, Pierre Ronceray, and Chase P. Broedersz

In this work the authors present a method to infer from image data the drift and diffusion, and hence the entropy production and time-resolved forces. They perform PCA on raw image data to identify components of independent variation. They filter components for eigenvalues that rise above a noise floor, whose relaxation time exceeds a threshold, and whose eigenvalues are clearly separated from those of lower-eigenvalue components. They then rank pairs of remaining components according to an approximation of their dissipation. On synthetic 2D data for two coupled beads or 5x5 nodes in a semiflexible filamentous network, they find that they can infer dissipation using significantly fewer components than for standard PCA, reasonably reconstruct time-resolved forces.

I generally like this paper. The problem is important and well-motivated, the method (despite being technically involved) is clearly presented, and the synthetic examples provide proof of principle for the method while spanning both a simple intuitive system and a more complex system closer to experimental situations of interest. So I am on the whole interested in seeing this paper appear in Nature Communications.

However, I feel that significant revisions are required to meet this bar. Briefly, I think the paper would be much strengthened by further discussion/exploration of:

- * the relative merits of this method compared to other methods;
- * the robustness of this method to variation of experimental noise and thermal fluctuations;
- * the applicability of this method in systems with less symmetry and more degrees of freedom than the synthetic examples;
- * the physical insights that might be gained by inference of time-resolved forces.

I expand on each of these points below, followed by suggestions that could help clarify the presentation, and finally a few possible typos.

3.1 Some comparison with other methods would be appreciated, for example Refs. [12] and [21], and the recent publication: [10.1103/PhysRevLett.124.120603/https://arxiv.org/abs/1910.00476](https://arxiv.org/abs/10.1103/PhysRevLett.124.120603).

In what contexts do you expect your method to exceed others? Are there entire classes of problems that are accessible / significantly more tractable due to your method? Does your method address entirely distinct questions from other methods?

3.2) There is little exploration of how the method performs as a function of noise magnitude. Moreover, it is unclear from the presentation how plausible the simulated parameters (including the noise magnitude) are.

3.3) Similarly, I see no discussion of how the method would be expected to perform with variation of the temperature, and hence the overall scale of thermal fluctuations. Fig. 3b shows data from $T_0 = 10^{-2}$, which is a very small temperature. Why is this chosen? Without more

detailed clarification, this would appear to strongly suppress fluctuations and thus ensure that the system is very nearly linear in its behavior.

3.4) SI section VI: how extensible is this method when the data does not have the regularities that you have in this example? I.e., is it sensitive to how you choose to partition your full image into patches? Is the uncertainty significantly magnified when you can't average your patches together because they have no resemblance to each other? Etc.

3.5) Some emphasis is placed on the importance of reducing dimensionality through PCA to a manageable number of degrees of freedom. In your chosen example, you can empirically find good eigenvalue separation between d eigenmodes and the rest. But the physical state space of any even moderately complex system will undoubtedly have d (proportional to the number of rigid components) much greater than $L \times W$. So how useful in practical situations would this actually be? And how does one choose the d to use when there is no clear separation like you see in your examples?

3.6) What is the advantage of being able to infer forces in image space? Can you suggest any insights this presents into the physical behavior of the system, that is not already captured by the principal components, which (to an untrained eye) look very similar to the instantaneous force maps?

MORE EXPLICIT ABOUT METHODS AND DATA DISPLAY

3.7) p6 mid right: "the dissipative components appear to reflect the local temperature inhomogeneities in the network". How do we see this in Fig 3d?

3.8) p6 mid right: "performing about twenty times better". 20x better by what measure?

3.9) Are dc 's just a reordering of pc 's? Do any dc 's appear more than once in the list? How do you jibe the fact that you choose dc 's by pairs, but then evaluate their effectiveness as individual dc 's?

3.10) Fig 3: could use some help interpreting the images of pc 's, dc 's, and forces. The characteristic paired black and red bars represent displacements of individual filaments (not filament vibrations since there are no filament degrees of freedom), and hence of (nearest-neighbor) pairs of nodes? Why (visually) are there no equivalent displacements in a given pc for the nodes?

3.11) Fig. S2c-d: What is the reason for the apparent increase of dissipation captured per pair of modes? Is it related to the combinatorial explosion of possible pairs of degrees of freedom to calculate A_{ij} , not because later pairs account for more dissipation "themselves"?

POSSIBLE TYPOS

* p1 bottom L: control → controls

* p3 top right: "may be need" → "may be needed"

* Fig 3 caption: I don't see any "arrows" in 3h.

- * p8 top right: "time traces of tracked object"
- * Paragraph including Eqs S2-S4: Seem to be inconsistent in usage of A vs A-dot?
- * pS3 bottom: "principled components"
- * pS4 bottom: loosing → losing

We would like to thank all referees for the time and effort spent on reviewing our paper, and their thoughtful and constructive feedback, which has sparked significant improvements of the manuscript.

Furthermore, we provide at the following link <https://www.dropbox.com/s/n844n6cpd5734e5/Code.zip?dl=0> a commented sample-version of the Python-code “Brownian_Movie_Analysis” used to produce the results of Fig. 3 of the manuscript. Please refer to the “README.txt” file for instructions on how to run the code. Sample output of the code is also provided.

We believe that the revised version of our manuscript is now suitable for publication in Nature Communications. We address all comments and questions point by point below. For convenience, the reviewers’ comments are shown in green in the response letter, and the corresponding changes in the revised manuscript are shown in blue.

Sincerely,
Chase Broedersz (LMU Munich and VU Amsterdam),
on behalf of all authors.

Reviewer # 1:

The authors set out to introduce a new data analysis technique for data obtained in microscopy experiments. In many soft matter experiments—especially with living matter—one obtains time-lapse microscopy videos of the systems evolution. Whereas, often people use data obtained from tracking embedded particles, the authors here aim to use the entire video for inference. Their goal is two-fold. One is to infer something about the entropy production of the system and two is to infer aspects of its dynamics.

Their method proceeds by lifting the dynamics to an image space where they estimate entropy production. They improve the data by processing it both using a PCA analysis as well as introduce a novel DCA (dissipation components analysis) to aid with the entropy production estimation. This is because, while the PCA is successful at finding directions where there is the most variation, they are not necessarily those directions that are most dissipative. Thus, the authors introduce another decomposition/projection of the dynamics using insights from linear diffusive processes.

The authors test their method on two examples. Two coupled beads in viscous media at two different temperatures as well as a network of coupled harmonic oscillators with a media of random temperatures. Their method is able to capture a sizable fraction of the true entropy production and in addition does a reasonable job of inferring the underlying force field. Both models are assumed to fulfill their assumption of Brownian dynamics

In many respects, this is a timely work. There has been a lot of interest lately in developing experimentally viable methods to infer entropy production in nonequilibrium matter, especially living systems. This paper brings to bear many of these tools and uses them in an innovative way.

We thank the reviewer for emphasizing the timeliness and the innovative aspects of our work.

1.1) My main reservation for this paper is whether it has broad applicability. The authors central assumption on which the method rests is that the observed physical system evolves according a multidimensional Brownian dynamics (Eq. 1). It is this assumption that allows them to use the formula in Eq. (2) to estimate the entropy production rate. Generically, one would expect that the degrees of freedom observed in a microscopy experiment on a complex nonequilibrium system are only part of all the dynamics occurring, and that many degrees of freedom are coarse-grained away. Thus, one would expect the observed dynamics to be highly nonlinear and non-Markovian. The question then arises is if you attempt to fit the dynamics to a Brownian dynamics (as in Eq. 1) and then infer the entropy production from the fitted dynamics (Eq. 2), does this entropy production have anything to do with true observable entropy production (accounting for non-Markovian and non-Gaussian effects)? It is my understanding that the answer here is false. There is no relation between the entropy production and the entropy production inferred (via Eq. 2) for a fit to a Brownian dynamics. This fact calls into question how principled this approach is, when one cannot be confident that the observed dynamics are well described by Eq. 1. I would suggest that the authors run their analysis on a non-Markovian/non-Gaussian example to test whether their estimated entropy production offers any relation to physical entropy production; and support those conclusions with analytic calculations.

We thank the reviewer for these comments. We first note that they concern not only our study, but also the numerous previous works where one considers dissipation in a system modeled by Brownian dynamics [such as reviewed in U. Seifert, “Stochastic Thermodynamics, Fluctuation Theorems and Molecular Machines”, Rep. Prog. Phys. 75, 126001, (2012), among many other references]. Indeed, Brownian dynamics is an effective dynamics obtained through coarse-graining fast, unobserved degrees of freedom (such as the solvent contributions), resulting in a dynamical noise that can in most cases be reasonably considered to be white and Gaussian. Within this model, it is well established that the steady-state entropy production rate, as quantified by our Eq. (2), $\dot{S} = \langle \mathbf{v} \mathbf{D}^{-1} \mathbf{v} \rangle$, corresponds to the steady-state heat dissipated into the bath (*i.e.* transferred to the fast degrees of freedom). This entropy production rate is thus widely studied and broadly accepted as an important quantification of the activity and dissipation in a system.

The reviewer is concerned that the entropy production rate we measure does not reflect the “true” entropy production. As we detail below, our measure provides a controlled lower bound to the true entropy production rate.

Schematically, there are three levels at which things can go wrong:

1. going from the all-atoms description of the physical system to a Brownian dynamics description of its slow degrees of freedom;
2. changing the level of coarse-graining of these degrees of freedom (for instance through imperfect imaging of the system);
3. performing the inference at a given level of coarse-graining.

The control of point (1) is the subject of stochastic thermodynamics; these are well accepted results, and questioning them is well beyond the scope of our article. We refer the reviewer to Ref. [39] for a discussion of the physical meaning of the entropy production rate quantified by Eq 2.

Point (2) is indeed of crucial importance for biological applications: typically, only a fraction of the degrees of freedom (such as the deformation of cytoskeletal filaments) will be observed by microscopy techniques. Instead of the “true” velocity and diffusion fields in the full high-dimensional phase-space, what is accessible to us are the average of these fields over unobserved degrees of freedom – which govern the dynamics of the observed degrees of freedom. This averaging over hidden degrees of freedom can also induce non-Markovian effects in the dynamics of the observed variables, which is not a problem for our inference method, as we explicitly discuss in the Main Text, Section I (lines 166-177).

However, one crucial property of the entropy production rate is that it can only decrease when coarse-graining the system: the fewer degrees of freedom are observed, the lower the measured dissipation. Mathematically, this stems from a convexity property of the local entropy production rate

$$s(\mathbf{x}) = \mathbf{v}(\mathbf{x}) \mathbf{D}^{-1}(\mathbf{x}) \mathbf{v}(\mathbf{x}) \equiv \pi(\mathbf{v}(\mathbf{x}), \mathbf{D}(\mathbf{x})) \quad (1)$$

at state \mathbf{x} . Indeed, the function $\pi(\mathbf{v}, \mathbf{D})$ thus defined is multivariate convex within the subspace of positive definite matrices \mathbf{D} . Note that positive-definiteness is a necessary property of the

diffusion matrix $\mathbf{D}(\mathbf{x})$ for every state \mathbf{x} . Given the convexity of π , Jensen's inequality implies:

$$\pi(\langle \mathbf{v} \rangle, \langle \mathbf{D} \rangle) \leq \langle \pi(\mathbf{v}, \mathbf{D}) \rangle \quad (2)$$

where $\langle \cdot \rangle$ stands for the averaging over any joint probability distribution $P(\mathbf{v}, \mathbf{D})$. This is in particular true when P represents the averaging over the possible values of the hidden degrees of freedom. The local entropy production rate $s(\mathbf{x})$ in the observable phase space is thus smaller or equal to the average entropy production rate of the full, high-dimensional phase space, conditioned on the value \mathbf{x} of the observation. We offer a proof of this point in the new Supplementary Sec. IX.

Finally, point (3) is discussed at length, through rigorous proofs, in the article introducing the inference method [Frishman and Ronceray, Phys. Rev. X 2020]. See in particular Fig. 4g for a numerical demonstration of this effect, Appendix D for a rigorous proof that the inferred entropy production is a lower bound to the actual one (modulo an error term that is kept under control self-consistently), and Appendix E for rigorous treatment of the non-Markovian effects of hidden variables.

In conclusion, our estimate is indeed a lower bound to the physical entropy production rate of interest, thereby addressing the concerns of the Referee, regarding the relevance of the quantity we measure. We now make this more clear in the Main Text, Section I (lines 166-177) and Supplementary Sec. IX.

We followed the reviewers suggestion to run the analysis on a non-Markovian example. In the revised Fig. 3 we now consider a more realistic and explicitly non-Markovian scenario, with hidden degrees of freedom due to a partial imaging of the system and due to limited optical resolution. We believe that these revisions help demonstrate the potential broader applicability of our approach.

1.2) Minor points:

1) Section III, after \hat{x} there is a stray period in the first paragraph.

Thanks for spotting this, we fixed the issue.

2) The authors also might find of interest (J. Phys. Conf. Series 750 (2016) 012003)) where those authors introduce and discuss the notion of probability angular momentum which bears a striking resemblance to the flux loops of the present article.

This is indeed a very relevant reference. We included a citation to this article in the second paragraph of the manuscript, after "the irreversible nature. . . ." (line 34).

3) In the author's analysis of the DCA in referring to figure 3d, they write "the dissipative components appear to reflect the local temperature inhomogeneities". Unfortunately, I cannot see this in Fig. 3d. While it's clear that the dissipative components have a weight distribution that is not evenly distributed, how does one make that connection to the variations in temperature?

We thank the referee for pointing out this unclear statement. The connection between temperature inhomogeneities and the DC's is not obvious. We changed this statement to

"The dissipative components exhibit a different spatial structure than the principal compo-

nents, as they aim to maximize different quantities”

Reviewer # 2 (Remarks to the Author):

Gnesotto et al. present a new approach for inferring non-equilibrium physical characteristics of systems, such as entropy production rate and time-resolved force distributions, from time-lapse microscopy imaging data. Their approach involves a new dimensionality reduction technique that uses dissipation to select a basis of modes. To test and demonstrate their approach, they use it to analyze the dynamics of two systems, a simple two-bead and spring system and a more complex filament network. The paper is well-written, with clear presentation of methodology and results, and presents a novel analysis approach that avoids the use of tracer particles and particle tracking. Due to the accessibility of the approach and its potential broad application to biophysics and soft-matter physics, the work represents a significant contribution to the suite of techniques for inferring the non-equilibrium physics of living and non-living systems.

We thank the referee for these supportive comments and for emphasizing the potential broad application of the application of our approach to biophysics and soft-matter physics.

While the manuscript is of good quality as is, I do have the following major and minor suggestions for strengthening the the work:

Major

2.1) While the authors demonstrate the effects of noise on their analysis approach, it would be interesting to know how robust the authors’ methods are to differences in optical resolution (or similarly, the size of the structure of interest relative to the optical resolution of the imaging system). Structure in biological systems, such as cytoskeletal elements of the cytoplasm, is often near or below the diffraction limit, and given that the manuscript is in part targeting experimentalists, addressing this practicality could strengthen the work. On a related note, is there some rationale for choosing the specific parameters used to generate the Brownian movies (e.g. Gaussian with variance 9 pixels in the case of the two-bead model)?

We thank the referee for their insightful suggestion. In this revised version, we have striven to demonstrate the robustness of our method to these effects. To this aim, we have introduced a new, more complex and realistic test system in Figure 3 of the main text: a randomly diluted, high temperature network, with only a small fraction of a large system visible in the movie. Specifically, we now include a quantitative study of the performance of our method as a function of the following parameters:

- Optical resolution (Gaussian blur parameter), presented in Figs. 3f and S4. The method consistently captures a sizable fraction of the dissipated entropy even when individual filaments cannot be resolved. Interestingly, a moderate level of blurring actually improves the resolution of the method, possibly because the image-space dynamics is expected to become more linear with a slightly blurred system. If blurring is further increased and objects in the movie start to overlap ($\sigma > 4$), the recovered entropy production rate starts to decline. Remarkably however, both PCA and DCA still yield non-zero estimates of the entropy production rate with very strong blurring $\sigma = 10$, if a sufficient number of

modes (> 10) are retained. Furthermore, for this example DCA outperforms PCA, also for blurred Brownian movies.

- Measurement noise in the pixel intensities (modeled as time- and space-uncorrelated fluctuations of intensity of each pixel), in Fig S5. The method is robust to noise levels comparable to the total intensity, leading to highly “scrambled” images.
- Finally, we vary the temperature scale (and thus the scale of the displacement of the filaments), and now show a high and low temperature scenario in the revised Fig 3.

2.2) How was the patch size for the patching procedure (results in Fig. 3F) chosen? Also, in Supplementary Fig. 4, it looks as though the centers of the patches lie at the vertices of the filament network. How do results depend on patch size and placement with respect to structure in the images? Again, addressing these questions could strengthen the work by accounting for practical considerations.

We realize that the procedure of tiling the system with patches that we described earlier was confusing and distracted from the main message we wanted to convey. We therefore removed this section in the revised manuscript. The key idea that makes the approach scalable is that useful information can be obtained about a system, even when only a smaller section of the system is observed. We now convey this point in the revised Figure 3, as detailed in our response to comment 2.4.

2.3) What are the reasons that PCA was chosen over other dimensionality reduction techniques. Non-negative matrix factorization, for example, would avoid negative pixel values (e.g. Fig. 2C). There are certainly good reasons to choose PCA, as evidenced by, for example, the methods the authors use to select the appropriate number of components for downstream analysis, but it would be nice to see some of this spelled out in the text.

We thank the reviewer for this comment. The precise dimensionality reduction method in the first part of our paper is not a particularly important aspect of our discussion; PCA is chosen as the simplest and most well-known method. The goal is to filter out the part of the data that is indistinguishable from the measurement noise, and bypass the “curse of dimensionality” through a generic representation of the dynamics in a lower-dimension space. Other reasonable generic dimensionality reduction methods could work just as well, although introducing additional nonlinearities in the analysis might result in less efficient entropy production inference in the subsequent treatment of the data.

One of the important propositions we make in this article is that a dimensionality reduction scheme *targeted at dissipative motion* is better adapted to entropy production inference. To this aim, we introduce Dissipative Component Analysis (DCA), obtained by diagonalizing the square of the Area Enclosing Rate matrix ($\dot{A} = \langle \dot{x}_i x_j - \dot{x}_j x_i \rangle$) to find the modes which dissipate the most. We believe that by presenting DCA alongside with the well-known PCA makes our procedure more comprehensible: entropy production inference is possible with both methods, but more efficient with the latter, as it targets the “right” modes. Overall, PCA is just one of possible ways of reducing dimensionality and eliminating noisy data, chosen to contrast with DCA; it should not be seen as an irreplaceable part of our methodology. Indeed, depending on the system that is studied other dimensional reduction schemes may be more effective. Finally, the negative values of the pixel intensities should not be seen as a concern: we are considering here deviations from the average pixel intensities, for which negative values are natural. How-

ever, we noted that the caption of Figure 2 was confusing, which we have now corrected.

To make our motivation for using PCA clear, we made the following changes in the main text:
“ To restrict ourselves to the manifold of images representing the physical states and to reduce the noise, we first perform a standard dimensionality reduction procedure: for simplicity, we employ Principal Component Analysis (PCA). As we shall see later, this standard procedure can be reinforced with an analysis that provides an additional basis transformation to select the most dissipative components. ”

And

“ We seek to reduce the dimensionality of the data and filter out measurement noise by finding relevant components. To this end, we employ Principal Component Analysis (PCA) [61] and determine the basis. . . ”

After

“Truncating the basis of dissipative components using the aforementioned criteria, allows us to identify a limited number of components that are assured to maximally contribute to the dissipation of the system. (line 455)”

We added:

“ This is analogous to PCA, where the diagonalization of the covariance matrix C allows one to select the components which capture most of the variance. ”

2.4) What are the reasons that the 5x5 filamentous network was chosen? In the introduction (last sentence), the authors argue that analysis of this network demonstrates that the approach is scalable to large systems, but I do not know that it is entirely clear given the scale and complexity of, for example, biopolymer networks in living systems. Additional justification or motivation for the particular system parameters could be sufficient to address this.

The comments of all three referees helped us realize that the 5×5 uniform filamentous network was not rich enough to serve as an illustrative example of the broader applicability of our approach to complex systems, such as biopolymer networks in living systems. To address this issue, we introduced the following changes in the system analyzed in the revised Fig. 3:

- We observe and analyze a smaller region of a much larger 20×20 network, addressing the typical situation where only a smaller section of a bigger experimental system is observed.
- We randomly dilute 20 % of the bonds in the network to introduce disorder and remove unrealistic spatial symmetries in the lattice.
- We strongly increase fluctuations by increasing the overall temperature of the system by a factor 5 w.r.t. the previous system shown in Fig. 3. We provide a thorough comparison between low and high temperature cases (see Fig. 3 of main text). To simplify the presentation, we now use a Bernoulli distribution of node temperatures.
- We blur the visible objects to simulate limited optical resolution and corrupt our movies with strong measurement noise (Supplementary Sec. VII,VIII)

Despite this increased complexity, our approach still manages to infer a significant amount of the entropy production rate in the observed part of the system, as shown in the revised Fig. 3. Furthermore, if we restrict ourselves to a smaller region of interest of the Brownian movie (see white frame in Fig. 3b - This region of interest has not been chosen ad hoc), we can still adequately reconstruct image forces (Fig. 3f-g). This force-inference procedure can be repeated on other regions of interest throughout the entire system. Because of this, and the stronger resemblance between the system analyzed here and an experimental one, we argue that our approach is scalable and applicable to realistic experimental biopolymer networks. In the revised manuscript, we made our motivation for the choice of this particular system more explicit.

2.5) Minor

A) Given the potentially broad readership, it could be nice to see a bit more motivation for why entropy production rate is such a quantity of interest.

We thank the reviewer for this suggestion. We have modified the introduction to clarify the role of the entropy production rate in active and biological matter, and what can be gained by measuring it:

“Such dissipative currents can be quantified by the entropy production rate [39], which is a measure of the irreversibility of the dynamics [40]. New approaches have been developed to measure the entropy production rate in real systems [22, 24], shedding light onto the structure of dissipative processes [19] and their impact on the dynamics of living matter [20]. However, it remains an outstanding challenge to accurately infer the entropy production rate by analyzing Brownian movies of such systems.”

B) Is there a reference for the claim that entropy production rate quantifies irreversibility (the sentence before Equation 2) and that irreversibility quantifies how far from equilibrium a system is (as implied by the preceding sentence)?

We have clarified these in the response to the previous point, and introduced corresponding references. Briefly:

- The link between entropy production rate and irreversibility is made particularly evident by Crook’s relation [Crooks Phys. Rev. E 60, 2721 (1999)], which relates entropy production along a trajectory, and the probability of observing the time-reversed trajectory. This reference is now included.
- The link between entropy production and how far from equilibrium a system is, is through the steady-state equality between entropy production rate and the heat dissipated in a thermal bath per unit time: entropy production and dissipation are equivalent in steady-state. These results, and their connection with the formulas we use to quantify entropy production, are well summarized in the classic review by Seifert [Seifert, Rep. Prog. Phys. 75 126001 (2012)].

C) Please add axes labels to the plots in Supplementary Fig. 3.

This Figure has been removed in the revised manuscript following point 3.5 of the third reviewer. Briefly, criterion 3) in our original manuscript will likely only be restrictive in the

special case of low-dimensional systems. Due to the specificity of criterion 3) to the system analyzed in the previous version of the manuscript, we have removed this criterion from the analysis and the related Supplementary Fig. 3 from our revised Supplementary Material.

D) Does DCA yield any new information or improvement to analysis of the two-bead system?

The two-bead system is a simple toy model, well suited for demonstrating the idea of inferring the entropy production rate directly from a Brownian movie. The simplicity of this model lies largely in the fact that it only has two degrees of freedom – the displacements of the two beads – the dynamics of which happen to be largely captured by the first two PCA components. DCA thus yields no further information, since PCA captures it all. This makes it different from most experimental systems, where the number of degrees of freedom will be much higher. In such a multidimensional system, DCA, designed to select the modes with most irreversible dynamics, can help to reduce the dimensionality of the system more effectively than, e.g., PCA, as demonstrated in the example of a filament network (revised Fig. 3). For the two-bead system already the first two PCA modes contain all information about the dynamics of the original system. Therefore, there is no need of reducing the dimensionality, and thus DCA cannot bring any improvement. Hence, we did not include the DCA analysis in Fig. 2.

E) On page 4, second column, second paragraph, is there a reference for the claim in the sentence beginning “Importantly, when inferring forces we always subtract from the drift...”

We added the following references to guide the reader to useful background literature on the topic of multiplicative noise and the related spurious drift:

- Risken, H. & Frank, T. The Fokker-Planck Equation: Methods of Solution and Applications. Springer Series in Synergetics (Springer-Verlag, Berlin Heidelberg, 1996), 2 edn.
- Lau, A. W. C. & Lubensky, T. C. State-dependent diffusion: Thermodynamic consistency and its path integral formulation. Phys. Rev. E 76, 011123 (2007).

F) In Fig. 2d, might there be a better way to illustrate the similarities between the exact and reconstructed images, such as some kind of highlighting of colocalization? It’s a bit hard to visually evaluate similarity as is.

We thank the referee for highlighting this issue, which we now address in Sec. III of the Supplementary Material. Here we compare the exact two-beads images \mathcal{I} to the reconstructed images \mathcal{I}_{rec} using PCA (see main text Fig. 2d). We do so with a scatter plot of the pixel values at different time points and with a kymograph of the difference between exact and reconstructed images, as shown in Supplementary Fig. 1. Overall, we find that the first four PCA modes allow for an accurate reconstruction of the images in the Brownian movie for this two-beads model.

G) Why was the particular model for imaging noise (uniformly sampled white noise) chosen? I suspect the particular noise model should not substantially affect results, but some combination of Gaussian white noise and shot noise (Poisson) might model imaging noise

in high resolution microscopy a bit more realistically (see e.g. <https://rupress.org/jcb/article/185/7/1135/35453/Accuracy-and-precision-in-quantitative>).

We thank the referee for bringing this important point to our attention. Indeed, we agree that white gaussian noise or shot noise are more realistic ways of modelling measurement noise in the imaging device. For this reason, we fully assessed the performance of our method at different levels of Gaussian white noise and with shot noise. The results are reported in the new Sec. VIII of the Supplement. The revised Fig. 3 of the main text refers now to a Brownian Movie with both image blurring and Gaussian white noise of amplitude $\alpha = 0.1$ (see Sec. VIII of the Supplement for more details). We believe that the new results demonstrating robustness to varying degrees of blurring help strengthen the broader applicability of our approach. More generally, this robustness stems from the robustness of Stochastic Force Inference (Frishman and Ronceray, PRX 2020), the method underlying the dynamical inference performed here, to measurement noise. This robustness is rigorously proven, and makes no assumption on the form of the noise, aside from it being time-uncorrelated.

Overall, the manuscript represents a significant contribution by introducing a new analysis approach that should be broadly of interest to those studying the non-equilibrium dynamics of soft matter and living systems. The clear presentation of results and methodology should facilitate adoption of the new analysis techniques. I would not want to unnecessarily delay publication of the work by burdening the authors with excessive additional work, but I hope that my comments might provide some constructive guidance for possible improvements.

We appreciate these comments on the significance of our contribution, the clarity of presentation, and we thank the reviewer for all their constructive suggestions, which substantially improved our manuscript.

Reviewer # 3 (Remarks to the Author):

“Learning the Non-Equilibrium Dynamics of Brownian Movies”

Federico S. Gnesotto, Grzegorz Gradziuk, Pierre Ronceray, and Chase P. Broedersz

In this work the authors present a method to infer from image data the drift and diffusion, and hence the entropy production and time-resolved forces. They perform PCA on raw image data to identify components of independent variation. They filter components for eigenvalues that rise above a noise floor, whose relaxation time exceeds a threshold, and whose eigenvalues are clearly separated from those of lower-eigenvalue components. They then rank pairs of remaining components according to an approximation of their dissipation. On synthetic 2D data for two coupled beads or 5x5 nodes in a semiflexible filamentous network, they find that they can infer dissipation using significantly fewer components than for standard PCA, reasonably reconstruct time-resolved forces.

I generally like this paper. The problem is important and well-motivated, the method (despite being technically involved) is clearly presented, and the synthetic examples provide proof of principle for the method while spanning both a simple intuitive system and a more complex system closer to experimental situations of interest. So I am on the whole interested

in seeing this paper appear in Nature Communications.

We thank the referee for these supportive comments regarding our manuscript and the importance of this problem.

However, I feel that significant revisions are required to meet this bar. Briefly, I think the paper would be much strengthened by further discussion/exploration of:

- * the relative merits of this method compared to other methods;**
- * the robustness of this method to variation of experimental noise and thermal fluctuations;**
- * the applicability of this method in systems with less symmetry and more degrees of freedom than the synthetic examples;**
- * the physical insights that might be gained by inference of time-resolved forces.**

We appreciate these constructive comments, which helped us substantially improve our manuscript. We now provide a much more realistic and generic example in the revised Figure 3 together with an expanded supplementary materials on issues such as blurring, measurement noise and the role of thermal fluctuations, which we believe addresses the main concerns of the reviewer. Also, we discuss in more detail the merits of our approach compared to other available methods and the insights obtained by inferring time-resolved forces.

I expand on each of these points below, followed by suggestions that could help clarify the presentation, and finally a few possible typos.

3.1 Some comparison with other methods would be appreciated, for example Refs. [12] and [21], and the recent publication: [10.1103/PhysRevLett.124.120603/https://arxiv.org/abs/1910.00476](https://arxiv.org/abs/1910.00476).

In what contexts do you expect your method to exceed others? Are there entire classes of problems that are accessible / significantly more tractable due to your method? Does your method address entirely distinct questions from other methods?

To best address the question, let us first highlight the main points of our procedure:

1. Input: time-lapse microscopy images (Brownian Movies)
2. Dimensionality reduction: finding the most dissipative components
3. Inference: estimating the entropy production rate and image forces in presence of measurement noise

The first point already makes our approach stand out. Most of the methods to date aimed at the inference of the entropy production rate, including the ones mentioned by the referee, take

as the starting point a set of time-trajectories. While such methods can be relevant for tracking-based experiments, there are cases where it is not obvious what to track, or where poor optical resolution does not even allow for a clear distinction of the degrees of freedom. Our method could be an alternative in such scenarios. In the revised manuscript we support this claim by analyzing the dynamics of a driven elastic network at varying levels of blurring, imitating different optical resolution. Even with extreme blurring (see Fig. 3f, $\sigma=10$) our method reliably detects positive entropy production rate under non-equilibrium conditions.

Whether tracking is possible or not, one of the remaining challenges in the inference problems for biological system is due to their typically large dimensionality. One way to overcome these difficulties is to develop methods capable of inferring the entropy production rate in high-dimensional systems. Examples of this direction of research include [Li et al, Nat. Commun. 10, 1666 (2019)] and its refined version [Manikandan et al, Phys. Rev. Lett. 124, 120603 (2020)]. There, one uses the Thermodynamic Uncertainty Relation to bypass the problem of accurately inferring the full high-dimensional dynamics, and to find a lower bound to the total entropy production rate. However, the dimensions considered in these papers ($d < 6$) are still far smaller than the dimensionality of many biological systems, such as cytoskeleton assemblies. This calls for a method of reducing the dimensionality of the problem, which preserves maximum information about the irreversibility of the observed dynamics. The 2nd point of our approach addresses this problem, allowing for the identification of the most dissipative components of the system. While it is exact only for linear system, we demonstrate that even with nonlinear dynamics it outperforms the standard Principal Component Analysis. Given the complexity of biological assemblies, our Dissipative Component Analysis may successfully complement trajectory-based Entropy Production Rate inference methods and improve their performance by selecting the right set of trajectories to analyze.

Having reduced the dimensionality, in the 3rd point we use the Stochastic Force Inference (SFI) [Frishman and Ronceray, Phys. Rev. X 10, 021009 (2020)] to find a lower bound to the total Entropy Production Rate. The advantage of using this particular method is that it gives a controlled error in presence of a measurement noise. Such noise, inevitable in any real data and imitated in our artificial experimental movies, is not considered in [Li et al, Nat. Commun. 10, 1666 (2019)] or [Manikandan et al, Phys. Rev. Lett. 124, 120603 (2020)]. Another advantage is that with no hidden degrees of freedom and a sufficiently large basis, SFI converges to the exact value of the entropy production rate. Meanwhile, methods based on the thermodynamic uncertainty relations, such as [Li et al, Nat. Commun. 10, 1666 (2019)], typically offer only a lower bound to the entropy production rate, even with a full access to the system's dynamics. This lower bound can be brought closer to the exact value by looking at shorter trajectories, as explained in [Manikandan et al, Phys. Rev. Lett. 124, 120603 (2020)]. These methods were developed to avoid estimating the force and mean velocity fields at every point of the multi-dimensional phase space. SFI circumvents this problem by expanding these fields in a basis and estimating only the coefficients in the expansion. Last but not least, the dynamics of image components may include spatially varying diffusion tensor, resulting in multiplicative noise. Such a scenario is not covered in [Li et al, Nat. Commun., 10, 1666 (2019)] or [Manikandan et al, Phys. Rev. Lett. 124, 120603 (2020)].

Finally, let us note that our approach would reveal no entropy production rate, if the apparent probability currents vanish due to averaging over the hidden degrees of freedom. This is in contrast to the approach presented in [Martinez et al, Nat. Commun. 10, 3542 (2019)] and based directly on the Kullback-Leibler divergence, which allows for detecting irreversibility

even in absence of probability currents. There, however, the considered set of states is discrete and it is not obvious to us how one could generalize this method to systems with a continuous set of states, as in our case.

3.2) There is little exploration of how the method performs as a function of noise magnitude. Moreover, it is unclear from the presentation how plausible the simulated parameters (including the noise magnitude) are.

Thanks for raising this issue. We now provide a thorough assessment of the performance of our method at different levels of blurring, Gaussian white noise and with shot noise. The results are reported in the new Sec. VIII of the Supplement. The revised Fig. 3 of the main text refers now to a Brownian Movie with Gaussian white noise of amplitude $\alpha = 0.1$ (see Sec. VIII of the Supplement for more details). We believe that the broader applicability of our approach is strengthened by these new results demonstrating robustness to varying degrees of measurement noise.

3.3) Similarly, I see no discussion of how the method would be expected to perform with variation of the temperature, and hence the overall scale of thermal fluctuations. Fig. 3b shows data from $T_0 = 10^{-2}$, which is a very small temperature. Why is this chosen? Without more detailed clarification, this would appear to strongly suppress fluctuations and thus ensure that the system is very nearly linear in its behavior.

We thank the referee for raising this important point. Indeed, with smaller temperatures, the system will be closer to linear and we should expect our approach to perform better. However, it is our aim to demonstrate that we can also provide insight into a more generic case with larger fluctuations. To this end, we significantly cranked up the temperatures of the nodes in the network example (now a considerably larger system with quenched disorder through bond dilution). The revised Fig. 3 of the main text presents a dataset where the temperature of the ‘cold’ nodes is increased by a factor 5 w.r.t. to the previous Fig. 3 (now $T_{\text{cold}} = 0.05$) and the temperature of the ‘hot’ nodes is increased by another factor of five ($T_{\text{hot}} = 0.25$). Note, to simplify the presentation, we now use a Bernoulli distribution of node temperatures. A movie of this example (Supplementary Movie 4) demonstrates that the system exhibits significant fluctuations.

Although this dataset is more challenging due to the presence of random bond-dilution, a large number of hidden degrees of freedom, and stronger fluctuations, our method is still able to recover a significant fraction of the entropy production rate (Fig. 3e) for the analyzed region of the network, and we can partially reconstruct image force field for a small region of interest (Fig.3 f-g). For comparison, we also analyzed a Brownian movie with lower temperatures ($T_{\text{cold}} = 0.01$, $T_{\text{hot}} = 0.05$). As expected for a system that is closer to linearity, our method captures a higher percentage of the total entropy production (Fig. 3e) and we can reconstruct the image forces to a higher level of precision (Fig. 3h).

3.4) SI section VI: how extensible is this method when the data does not have the regularities that you have in this example? I.e., is it sensitive to how you choose to partition your full image into patches? Is the uncertainty significantly magnified when you can’t average your patches together because they have no resemblance to each other? Etc.

We would like to stress that we did not average patches together: panel i) of the original Fig. 3 included force-inference data from all disjoint patches in which the network is subdivided. Put

simply, the idea was that one can use a tiling of smaller regions of interest to reconstruct force inference data of a large system. However, we realize that the procedure of tiling the system with patches that we described earlier was confusing and distracted from the main message we wanted to convey. We therefore removed this section in the revised manuscript. The key idea that makes the approach scalable is that useful information can be obtained about a large assembly, even when we only a smaller section of the system is observed.

The comments of all three reviewers helped us realize that the 5x5 uniform filamentous network was too sterile to serve as an illustrative example of the broader applicability of our approach to complex systems, such as biopolymer networks in living systems. To address this issue, we introduced the following changes in the system analyzed in the revised Fig. 3:

- We observe and analyze a smaller region of a much larger 20x20 network, addressing the typical situation where only a smaller section of a bigger experimental system is observed.
- We randomly dilute 20% of the bonds in the network to introduce disorder and remove unrealistic spatial symmetries in the lattice.
- We strongly increase fluctuations by increasing the overall temperature of the system by a factor 5 w.r.t. the previous system shown in Fig. 3. We provide a thorough comparison between low and high temperature cases (see revised Fig. 3). To simplify the presentation, we now use a Bernoulli distribution of node temperatures.
- We blur the visible object to simulate limited optical resolution and corrupt our movies with strong measurement noise (Supplementary Sec. VII,VIII).

Despite this increase complexity, our method still manages to infer a significant amount of the entropy production rate in the observed part of the system, as shown in the revised Fig. 3. Furthermore, if we restrict ourselves to a smaller region of interest of the Brownian movie (see white frame in Fig. 3b), we can still adequately reconstruct image forces (Fig. 3f-h). Note, this region of interest has not been chosen ad hoc. This force-inference procedure can, in principle, be repeated on other regions of interest throughout the entire system, and therefore we argue that our approach is scalable and applicable to realistic experimental biopolymer networks. In the revised manuscript, we made our motivation for the choice of this particular system more explicit.

3.5) Some emphasis is placed on the importance of reducing dimensionality through PCA to a manageable number of degrees of freedom. In your chosen example, you can empirically find good eigenvalue separation between d eigenmodes and the rest. But the physical state space of any even moderately complex system will undoubtedly have d (proportional to the number of rigid components) much greater than $L \times W$. So how useful in practical situations would this actually be? And how does one choose the d to use when there is no clear separation like you see in your examples?

Thank you for raising this issue. It is true that in a generic biological system the number of the degrees of freedom is likely to be very high. In the case where no clear gap appears in

the eigenvalues spectrum, the 3rd criterion (in our original manuscript) would not yield any threshold and is therefore non-restrictive. In this case, the set of components needs to be truncated based on the first two criteria. Note, however, the high/infinite dimensionality of a physical system does not necessarily imply that the set of corresponding images will be equally high-dimensional. Given the finite resolution of the images, many of the physical degrees of freedom may get lost (hidden) in the imaging process, and the number of physical degrees of freedom that enter the images may be significantly decreased. An example would be a filament of length L imaged with resolution r . Despite this filament having infinitely many bending modes, only L/r of them can be recovered from the image. In this case, criterion 3 may be restrictive. Nevertheless, for a system as complex as a polymer assembly (such as the example in revised Figure 3), the number of physical degrees of freedom that are resolved by the images can still exceed the number of modes that can be included in the inference procedure, given the computational complexity of performing stochastic inference in high-dimensional spaces. We believe the 3rd criterion to be most useful in systems in which the imaged degrees of freedom are truly low dimensional, e.g. a couple of visible beads inside an invisible gel, similar in spirit to the example in Fig. 2. In this example, criterion 3 is restrictive as it leads to a truncation at 2 PC's, which is important for the accurate estimate of the Entropy Production Rate, as pointed out in the discussion of the results in the main text. Admittedly though, this is a special case. Therefore, we decided not to include criterion 3 as an essential part of the procedure and only mention it as complementary to criteria 1 and 2, for low dimensional systems. We expressed this in an extended discussion of the two-bead system (line 325):

“Note that including more modes than the dimension of the physical phase space (in this case 2) can lead to an overestimate of \dot{S} (Fig. 2g).” “ In such low-dimensional systems, one can further restrict the number of included components based on estimating the dimensionality of the set of images in the Brownian movie. ”

The previous discussion of criterion 3 has been removed from the Supplementary material.

3.6) What is the advantage of being able to infer forces in image space? Can you suggest any insights this presents into the physical behavior of the system, that is not already captured by the principal components, which (to an untrained eye) look very similar to the instantaneous force maps?

The principal components are a property of the variability of the collections of images in the Brownian movie, and they contain no direct information about the underlying dynamics. By contrast, the force fields in image space relies on performing Stochastic Force Inference. The resulting force fields provide insight into the spatial structure of the instantaneous deterministic forces in the system at a given configuration, reflecting the underlying deterministic forces that evolve the recorded system. Intuitively, the inferred forces indicate the “force” acting on each pixel, *i.e.* whether its intensity value is prone to rise or fall at the next time step. We included the following statement in the discussion of force fields in the context of Figure 3 (line 488)

“These force fields provide insight into the spatial structure of the instantaneous deterministic forces in the system at a given configuration. In image space, these forces describe the dynamics of the pixel: positive and negative image forces represent a deterministic force acting to respectively raise and lower pixel values, which reflect the forces acting on the position and shape of the objects being imaged.”

MORE EXPLICIT ABOUT METHODS AND DATA DISPLAY

3.7) p6 mid right: “the dissipative components appear to reflect the local temperature inhomogeneities in the network”. How do we see this in Fig 3d?

We thank the referee for pointing out this unclear statement, which has now been changed to:

“The dissipative components exhibit a different spatial structure than the principal components, as they aim to maximize different quantities”

3.8) p6 mid right: “performing about twenty times better”. 20x better by what measure?

We agree that this statement about the fraction of entropy production captured was too vague, and we replaced it with

“Strikingly, DCA allows us to recover a larger portion of the entropy production rate of the observed region (almost 10% with 30 components), performing consistently better than the PCA-based approach, as shown in Fig. 3e”

3.9) Are dc’s just a reordering of pc’s? Do any dc’s appear more than once in the list? How do you jibe the fact that you choose dc’s by pairs, but then evaluate their effectiveness as individual dc’s?

The DCA modes are not just a reordering of the PCA modes. To find the PCA modes one simply has to diagonalize the covariance matrix, without inputting any information about the time-ordering of the data. On the other hand, the DCA modes are constructed by first rescaling the PCA modes to ensure that the covariance matrix equals identity, and then performing a rotation which brings the square of the Area Enclosing Rate matrix to a diagonal form – the latter containing crucial information about the dynamics. There is no reason in general for these two bases to coincide. In the most general case, the DCA modes can be obtained from the PCA modes, by rescaling each mode independently and performing an orthonormal transformation. Therefore, each DCA mode may be composed of arbitrarily many PCA modes.

While the PCA modes capture the degrees of freedom with the highest variance, the DCA modes are designed to capture the irreversible dynamics so one can expect them to have a qualitatively different spatial structure.

When ordering the DCA modes, we always look at them in pairs and order them according to the estimated contribution of each pair to the total entropy production rate. Inspired by the referee’s questions, we replaced the following, possibly confusing sentence:

“ The idea behind this approach is that the components are hierarchically ordered according to how much they contribute to the entropy production, such that it becomes possible to truncate the basis and reduce the dimensionality of the problem, while retaining maximum information about the system’s irreversibility. ”

with (line 197)

“ The idea behind this approach is to find an appropriate basis, in which pairs of components can be hierarchically ordered according to how much they contribute to the total entropy production rate. It then becomes possible to truncate the basis and reduce the dimensionality of the problem, while retaining maximum information about the system’s irreversible dynamics. ”

3.10) Fig 3: could use some help interpreting the images of pc's, dc's, and forces. The characteristic paired black and red bars represent displacements of individual filaments (not filament vibrations since there are no filament degrees of freedom), and hence of (nearest-neighbor) pairs of nodes? Why (visually) are there no equivalent displacements in a given pc for the nodes?

We thank the referee for raising this point. This issue was an artifact of the high symmetry of our the uniform network in our previous version. Briefly, when generating the movies, for a given configuration we only plot the bonds connecting the nodes. The determination of both PCA and DCA modes is based on the variations of pixel intensities. In particular, regions with no or little intensity variations are less likely to be pronounced in the leading PCA or DCA modes. Note that given the way we image the network, the nodes were surrounded in all lattice directions by bright filaments (previous version of Fig. 3). Thus, when a node was displaced, there was typically an end of one of the bonds appearing in its place and thus no significant difference in the intensity of the corresponding pixels could be observed. Consequently, the regions corresponding to the rest positions of the nodes appeared as white in the leading PCA and DCA modes. In case of the bonds, any displacement perpendicular to the bond direction results in a significant change of the pixel intensities. This is why the displacements of the bonds were pronounced in the leading PCA and DCA modes much more than the displacements of individual nodes.

Note that in the revised manuscript we consider a randomized network at a significantly higher temperature (revised Fig. 3). This constitutes a more generic case than the previously considered system. As a result, the nodes are no longer surrounded by the bonds from all sides (along the lattice directions) and their displacements are much larger than in the old version. Therefore, the issue discussed above is no longer present in the new version of the model.

3.11) Fig. S2c-d: What is the reason for the apparent increase of dissipation captured per pair of modes? Is it related to the combinatorial explosion of possible pairs of degrees of freedom to calculate A_{ij} , not because later pairs account for more dissipation “themselves”?

Unfortunately, Fig. S2 in the original manuscript illustrates the truncation criteria and seems unrelated to the reviewer's question. We believe that the question refers to the plots of the inferred Entropy Production Rate (EPR) as a function of the number of included modes, as the ones in Fig. 3. There one can see the typically convex plots for the PCA modes (at least in the initial range) and concave plots for the DCA modes.

The convexity of the PCA-plots could at first glance suggest that the consecutive modes themselves contribute more and more to the EPR. This is, however, incorrect: one cannot in general assign an EPR to a single mode or even a pair of modes. This can be understood by looking at Eq. (4), where the total EPR is expressed in terms of the mean phase space velocity $v(x)$. When adding a new mode to the set of already included modes, the inferred EPR increases not only by the contributions from the new mode, but also due to more detailed information about the $v(x)$ field of the old modes (less averaging over hidden variables – see the discussion after Eq. 4). Therefore, inclusion of every new mode can increase the contributions from all the old modes. This can explain the accelerated growth of the EPR with the number of included PCA modes.

The situation changes for the DCA modes. These modes are constructed such that, for a linear

system, the mean phase space velocity of each DCA-pair is independent of all the other modes. Therefore, by observing a single pair of DCA modes we can infer the complete $v_p(x)$ field of that pair and no additional structure is uncovered by including more modes. This can be expressed as: $\langle v_p(x) | x_p \rangle = v_p(x)$ for all x , where x_p is a vector of coefficients of the selected pair of DCA modes. Consequently, adding a new pair to the set of already included modes doesn't change significantly the contributions to the EPR from the old modes and the inferred EPR changes mostly by the contribution from the new pair. After ordering the DCA pairs according to their predicted contribution to the total EPR we obtain the concave plots where the consecutive increments of the inferred EPR are decreasing.

POSSIBLE TYPOS

- * p1 bottom L: control \rightarrow controls
- * p3 top right: "may be need" \rightarrow "may be needed"
- * Fig 3 caption: I don't see any "arrows" in 3h.
- * p8 top right: "time traces of tracked object"
- * Paragraph including Eqs S2-S4: Seem to be inconsistent in usage of A vs A-dot?
- * pS3 bottom: "principled components"
- * pS4 bottom: loosing \rightarrow losing

Thank you for bringing all these typos to our attention. They have all been addressed in the revised version of the manuscript. In particular, we now consistently use only A-dot for the area enclosing rate.

REVIEWERS' COMMENTS:

Reviewer # 1 (Remarks to the Author):

I am impressed with the latest version of the manuscript. They have thoughtful and carefully replied to comments. The new example is a much more convincing illustration of the method they propose. I imagine that many researches will be employing it in the near future. With that, I can offer my recommendation for publication.

Reviewer # 2 (Remarks to the Author):

The major claims of the paper remain unchanged from the initial submission, and I stand by the general tenor of my initial comments. Briefly, the work represents a sound and significant contribution to the study of the physics of non-equilibrium systems. I find the response to the reviewer comments to be thorough and sufficient and that the paper has been substantially improved in the course of addressing the comments. In particular, Figure 3 and the corresponding analysis is now much more compelling and supports the authors' claims that the techniques are scalable and applicable to experimental systems. By increasing the complexity of the filament network and expanding their exploration of robustness in the face of more realistic noise profiles, the authors have demonstrated how their analysis methods might perform when faced with the complexity of experimental systems.

Reviewer # 3 (Remarks to the Author):

I strongly urge publication in Nature Communications of this revised manuscript. The authors have done an admirable job of addressing (often with quite substantial additions) my and the other reviewers' extensive comments, in the process significantly improving the clarity of the presentation and the robustness of the results. I have no major concerns remaining.

RESPONSE:

We thank all three referees for their time and efforts in reviewing our paper, and for their previous constructive suggestions. All reviewers are satisfied with our previous response and revisions, and they recommend publication.